# Effects of brown coatings on the absorption enhancement of black carbon: a numerical investigation

Jie Luo, Yongming Zhang, Feng Wang, and Qixing Zhang

State Key Laboratory of Fire Science, University of Science and Technology of China, Hefei, Anhui 230026, China

**Correspondence:** Qixing Zhang (qixing@ustc.edu.cn)

**Abstract.** Using the numerically exact multiple sphere T-matrix (MSTM) method, we explored the effects of brown coatings on absorption enhancement ($E_{abs}$) of black carbon (BC) at different wavelengths ($\lambda$). In addition, the ratio of the absorption of BC coated by brown carbon (BrC) to an external mixture of BrC and BC ($E_{abs\_internal}$) is also investigated. In this work, thinly-coated BC is defined as those with BC volume fraction over 20%, and other BC is considered to be thickly-coated. $E_{abs}$ increases with the absorption of coatings, while an opposite trend is observed for $E_{abs\_internal}$. A much wider range of $E_{abs}$ is observed for BC with brown coatings compared to that with non-absorbing coatings. As the mass ratio of BrC to BC increases to 13.9, $E_{abs}$ can reach approximately 5.4 for BC with brown coatings at $\lambda = 0.35\ um$ under a typical size distribution. Previous studies have focused on the lensing effects of coatings but neglected the blocking effects of absorbing coatings. $E_{abs\_internal}$ can be below 1 at ultraviolet spectral region for BC with brown coatings, which indicates that the absorption of internally mixed BC is less than that of an external mixture of BrC and BC due to the blocking effects of outer coatings, and we named the blocking effect of absorbing coatings "sunglasses effect". In addition, the applicability of core-shell sphere model are also evaluated for BC with brown coatings. Absorption cross-section ($C_{abs}$) of thickly-coated BC is underestimated by core-shell sphere model for all wavelengths while the underestimation becomes negligible as the imaginary part of the refractive index of brown carbon ($k_{BrC}$) turns very large. The lensing effect and the sunglass effect is clearly defined. Moreover, the effects of composition ratios, the size distribution are explored at different wavelengths. Our findings can improve the understanding of the absorption enhancement of BC with brown coatings.

## 1 Introduction

Recent modeling and field studies have indicated that aerosol light absorption is an important contributor to climate forcing (Jacobson, 2001; Krishnan and Ramanathan, 2002; Bond et al., 2013). Black carbon (BC) , which is a product of incomplete combustion, is the strongest solar-absorbing aerosol in the atmosphere (Lack et al., 2009; Zhang et al., 2008b). BC radiative forcing from fossil fuels and biomass burning have been estimated to be approximately $0.4 W/m^2$, as the second anthropogenic contributor (after $CO_2$) to climate forcing due to their strong absorption of solar radiation (Forster et al., 2007; Schwarz et al., 2008). Sensitivity tests suggest that the mixing state and morphology of BC aerosols can largely affect the absorption of BC (Ma et al., 2012; Zhang et al., 2018). Due to the large uncertainties of BC morphologies and mixing states, the understanding of BC absorption is still limited. Even when coated with non-absorbing materials, the BC absorption can be enhanced (Liu

et al., 2017; Cappa et al., 2012). Many studies mainly attribute the absorption enhancements ($E_{abs}$) to the lensing effect (Bond et al., 2006; Fuller et al., 1999b).

For the estimation of BC absorption enhancements, many field measurements have been conducted. Naoe et al. (2009) presented factors of 1.1-1.4 for BC absorption enhancement at a suburban site in Japan, while Cui et al. (2016) indicated that the absorption enhancement factors increase from $1.4\pm0.3$ during fresh combustions to $\sim 3$ for aged BC in a rural site over the North China Plain (NCP). Liu et al. (2017) found that BC absorption enhancement is significantly influenced by the particle mixing state. The measured range of $E_{abs}$ is approximately $1\sim 1.5$. You et al. (2016) observed the wavelength-dependent absorption enhancement of coated BC. In their measurements, $E_{abs}$ increased up to 3 at the shortest measured wavelengths, while it was approximately 1.6 in the near-IR wavelength. A negligible absorption enhancement of only 6% for ambient BC particles was reported by Cappa et al. (2012) based on direct measurements over California (USA). Chen et al. (2017) reported an average $E_{abs}$ of $2.07\pm0.72$ for the urban haze in winter in northern China. However, this result was time-dependent. The absorption enhancement of BC during the urban PM2.5 pollution was $1.31\pm0.29$ in the morning, while in the afternoon, it increased to approximately $2.23\pm1.05$; then, it decreased to $1.52\pm0.75$ in the evening. Recently, larger $E_{abs}$ value of 2.6-4.0 at Beijing, China was reported by Xu et al. (2016). In summary, the reported $E_{abs}$ values are not consistent in different studies due to the complex aging statuses.

Although the field measurements can provide referential absorption enhancement values for different aging statuses and regions, causes of these enhancements are not clear. For example, what is the main factor that causes the complex absorption enhancements: morphology, the mixing states or the types of coatings? To our best knowledge, field measurements have difficulty answering these questions currently. Numerical simulation is a strong tool that reveals the mechanism responsible for the complex absorption enhancements. To improve the understanding of the complex absorption enhancements of BC, numerical studies have also been conducted. For instance, based on the core-shell Mie theory, the absorption enhancement factors have been estimated up to 3 (Bond et al., 2006). By the numerically exact multiple sphere T-matrix (MSTM) method, Zhang et al. (2017) presented the absorption enhancements of non-absorbing coatings for aged BC ranging from 1.1-2.4, and they were significantly influenced by the morphology and aging statuses but insensitive to the BC refractive index. However, previous studies have failed to uncover the effects of coating absorption. In their studies, coatings were considered as non-absorbtive, and BC absorption enhancements were completely caused by lensing effects. Nevertheless, in the atmosphere, there is a type of organic carbon (OC) that absorbs the radiation in the range of the ultraviolet and visible spectra, which is known well as brown carbon (BrC); BC can also be mixed with BrC. Compared with non-absorbing materials, the absorption of BrC is significantly wavelength-dependent, and the imaginary part of the refractive index for BrC has a wide range (Kirchstetter et al., 2004), which results in large uncertainties for the estimation of aerosol absorption. Therefore, the absorption of BrC has gained increasing interests (Kirchstetter et al., 2004; Shamjad et al., 2018).

Many studies have been conducted to evaluate the absorption of BrC. One typical method for the determination of BrC absorption is isolating BrC by extracting filtered samples (Cheng et al., 2017). This method can be used to determine the imaginary part of the BrC refractive index. However, it is difficult to understand the effects of BrC on the total aerosol absorption, as BrC is commonly mixed with other chemical compositions. The assumption of externally mixing can be used to evaluate

the absorption of BrC and BC separately. Nevertheless, in many cases, BC is internally mixed with other materials. It is widely accepted that the absorption is underestimated by the external mixing assumption when BC is coated with non-absorbing materials due to lensing effects. However, whether this is true for BC with BrC coatings is not clear. To understand the effects of BrC coatings, the contributions of "lensing effects" and the total absorption enhancement of BC with BrC coatings should be analyzed individually.

Cheng et al. (2017) has conducted a numerical investigation on BC absorption enhancement, BrC absorption enhancement and lensing effects on BC mixed BrC by assuming a core-shell structure. While the internal mixing of BC is widely accepted, the core-shell structure is debated (Adachi et al., 2010; Cappa et al., 2012; Bond et al., 2013). He et al. (2015) developed a theoretical BC aging model and concluded that the evolution of coating thickness, morphology, and composition during the aging process could have significant impacts on BC absorption. Freshly emitted BC commonly presents fractal structures. As the BC ages in the atmosphere, BC becomes more compact and OC materials can condense onto the particles. Therefore, BC can be embedded in an OC shell (China et al., 2013; Wang et al., 2017). When non-BC fraction is low, BC can still present near fractal structure (referred as thinly-coated BC in this study) (Wang et al., 2017). As BC is further coated, BC aggregates are collapsed into more compact and spherical clusters when fully engulfed in coating material (referred as thickly-coated BC in this study) (Coz and Leck, 2011a; Zhang et al., 2008a).

In this study, a numerical investigation was conducted to explore the factors that contribute the complex absorption enhancement of BC with BrC coatings for different mixing states. Two types of mixing states were considered: thinly-coated BC and thickly-coated BC. Thinly-coated BC is assumed as those with BC volume fraction over 20%, and other BC is considered to be thickly-coated. The results would give further understanding for the causes of BC absorption enhancements and suggestion for the inferred BC mixing states.

## 2 Methodology

### 2.1 Geometric properties of BC aerosols

In climate modeling, a spherical shape is commonly assumed for aerosols and can be calculated with high efficiency using the Mie theory (Mie, 1908). However, in many cases, this shape can introduce large errors compared with the measurements due to the oversimplification of the shape. Recently, the nonsphericity of aerosols has gained increasing interests (Yang et al., 2003; Bi and Yang, 2016). Specifically, observations have indicated that uncoated BC particles are commonly composed of numerous small spherical particles. Fractal aggregates can be greatly used to describe their geometric properties. Mathematically, the structure satisfies the well-known fractal law (Mishchenko et al., 2002):

$$n_s = k_0 (\frac{R_g}{R})^{D_f} \tag{1}$$

$$R_g^2 = \frac{1}{n_s} \sum_{i=1}^{n_s} l_i^2 \tag{2}$$

where $n_s$ represents the number of the monomers in the cluster, $R$ represents the mean radius of the monomers, $k_0$ represents the fractal prefactor, $D_f$ represents the fractal dimension, $R_g$ represents the radius of gyration, and $l_i$ represents the distance from the $i$th monomer to the center of the cluster.

The fractal dimension is a key parameter that describes the compactness of BC aggregates (Sorensen, 2001; Sorensen and Roberts, 1997; Luo et al., 2018a). Generally, aggregates tend to be more compact with the increase in $D_f$. A $D_f$ of 1 can describe an open-chain-type shape, while the aggregates tend to be spherical as $D_f$ approaches 3. Numerous experimental studies have been carried out to evaluate the $D_f$ of BC aggregates. Immediately after they are emitted, BC aggregates generally exhibit fluffy structures with a small fractal dimension ($D_f$), that is normally less than 2, such as the $D_f$ of BC aggregates from biomass burnings (1.67-1.83) (Chakrabarty et al., 2006), the $D_f$ of BC from vehicle emissions (1.52-1.94) (China et al., 2014), and the $D_f$ of BC from diesel combustion (1.6-1.9) (Wentzel et al., 2003).

However, under the effects of atmospheric aging, the structures and chemical compositions of BC may change. Aged BC tends to be mixed with other chemical components, and the shape becomes more compact. Therefore, in the atmosphere, aggregates can have fractal dimensions up to 2.6 (Chakrabarty et al., 2006). In some cases, BC aggregates are thinly coated with other materials, and still exhibit a fractal structure. However, different from freshly emitted BC aggregates, both lacy and compact structures can exist. Therefore, for thinly-coated BC, the $D_f$ was assumed to be in the range from $1.8 - 2.6$. As BC becomes increasingly coated, BC aggregates may transform from highly agglomerated to nearly spherical particles. A $D_f = 2.6$ was assumed for thickly-coated BC. Even though a fractal prefactor can also vary under different combustion and aging statuses, it has less significant effects on the absorption of BC compared to the $D_f$. When fixing $D_f$ to be 1.82, Liu and Mishchenko (2005) demonstrated that the absorption cross-section of BC aggregates does not change substantially as fractal prefactor varies from 0.9 to 2.1. Therefore, a fixed fractal prefactor of 1.2 was assumed in this work.

The monomer radius and monomers number are two key parameters that determine the particle size. Even though the monomers radii are polydispersed in the atmosphere, they vary within a narrow range. Monomer radii are commonly observed within $\sim$ 10-25 $nm$ (Bond and Bergstrom, 2006). In addition, Kahnert (2010b) demonstrated that $C_{abs}$ is insensitive to monomer radii when the monomer radii are within $\sim$ 10-25 $nm$. As a result, for convenient application, a fixed monomer radius of $R =$20 $nm$ was assumed in this work. Based on TEM/SEM image, the monomer number $n_s$ can reach approximately 800 (Adachi and Buseck, 2008). Values of $1 \leq n_s \leq 1000$ were considered in this work. For an aggregate with $n_s$ monomers, the equivalent radius was given by the equivalent volume sphere radius $R\sqrt[3]{n_s}$. The morphological parameters considered in this work are shown in Table 1.

## 2.2 Generation of BC aerosols

The morphologies of coated BC considered in this work are classified into two categories: thinly-coated BC and thickly-coated BC. The closed-cell structure, which is an example of where coating material that not only covers the outer layers of BC aggregates but also fills the internal voids among primary spherules, can be used to represent the thinly-coated BC (Liou et al., 2011; Strawa et al., 1999). In addition, Kahnert (2017) demonstrated that the absorption of closed-cell structures and more realistic morphologies do not have large deviations. Therefore, it is reasonable to use the closed-cell model for calculating the

absorption of thinly-coated BC, while the thickly-coated BC are commonly represented by a structure where BC aggregates are encapsulated in a sphere (Zhang et al., 2017; Cheng et al., 2014). The typical morphologies are shown in Figure 1.

Diffusion-limited algorithms (DLA), including the particle-cluster aggregation (PCA) (Hentschel, 1984) and the cluster-cluster aggregation (CCA) methods (Thouy and Jullien, 1994), have been developed for the generation of aggregates. However, adjustable DLA codes are commonly applied due to its quick implementation and adjustable fractal parameters (Koylu et al., 1995). In this work, an adjustable DLA code developed by Woźniak (2012) was used. Compared with ordinary DLA codes, this code preserves fractal parameters during each step of the aggregation, which avoids the generation of multifractal aggregates (Jensen et al., 2002). After the generation of the aggregates, the coatings were added. More specifically, for thinly-coated BC, the BrC shells were generated by the adjustable algorithm, and then the BC cores were added; the details are shown in previous studies (Luo et al., 2018c; Wu et al., 2014). The thickly-coated BC is generated by covering the BrC spherical coatings on the BC aggregates, as shown in the study of Cheng et al. (2015).

## 2.3 Light scattering method

To calculate the radiative properties of BC in this work, numerical solution methods from Maxwell's equations, including the finite-difference time-domain (FDTD) method (Yee, 1966; Taflove and Hagness, 2005), generalized multiparticle Mie (GMM) method (Xu, 1997; Xu and Gustafson, 2001), numerically exact multiple sphere T-matrix (MSTM) method (Mackowski and Mishchenko, 2011; Mishchenko et al., 2004), the geometric-optics surface-wave (GOS) method (Liou et al., 2011; He et al., 2016) and discrete-dipole approximation (DDA) method (Draine and Flatau, 1994; Laczik, 1996; Yurkin and Hoekstra, 2007), can all be used. However, compared with other numerical methods, the MSTM has an advantage for the calculation of optical properties for randomly oriented particles analytically without numerically averaging over particle orientations. Therefore, this method has high efficiency to calculate optical properties of BC. In this work, the latest MSTM code, MSTM version 3.0 (Mackowski, 2013), was applied.

In this study, all the radiative properties of BC were calculated based on the assumption that BC particles and their mirror counterparts are present in equal numbers in ensemble of randomly oriented particles. In the atmosphere, it is reasonable to assume that the possibility of each particle direction is identical, which mathematically satisfies the definition of random orientation (Mishchenko and Yurkin, 2017).

## 2.4 Calculating absorption enhancement of BC

The presence of non-BC coated materials can result in the enhancement of BC absorption, referred as BC absorption enhancement ($E_{abs}$). Therefore, $E_{abs}$ can be defined as the amplification of BC absorption after BC being coated:

$$E_{abs} = \frac{C_{abs\_coated}}{C_{abs\_bare}} \qquad (3)$$

where $C_{abs\_coated}$ and $C_{abs\_bare}$ represent the absorption cross-sections of coated BC and bare BC respectively.

As BrC also absorbs solar radiation, it is also desirable to compare the absorption of BC coated by BrC coatings with BC and an external mixture of BrC and BC. The absorption of BrC shell is calculated as:

$$C_{abs\_BrC\_shell} = C_{abs\_BrC(\ coated\ shape)} - C_{abs\_BrC(\ bare\ shape)} \tag{4}$$

where $C_{abs\_BrC(\ coated\ shape)}$ and $C_{abs\_BrC(\ bare\ shape)}$ represent the absorption cross-sections of BrC with morphologies that are identical to coated BC and bare BC, respectively. The calculation of the absorption of BrC shell is shown in Figure S1. In this process, we assume that the absorption of BrC with the same shape as the coated BC is identical to the external mixture of BrC that has the same shape as bare BC and BrC shell. We must clarify that this disposal method neglects the blocking effect and lensing effect of outer BrC shell on the internal BrC. However, as the BrC absorption is significantly less than the BC absorption with identical shape, the absorption caused by the blocking effect and lensing effect of outer BrC on the internal BrC is relative small compared with the BC absorption. Therefore, it is reasonable to make some simplifications.

In this work, we defined an parameter ($E_{abs\_internal}$) to represent the ratio between the absorption of BC coated by BrC coatings and an external mixture of BrC and BC:

$$E_{abs\_internal} = \frac{C_{abs\_coated}}{C_{abs\_BrC\_shell} + C_{abs\_bare}} \tag{5}$$

## 2.5   Size distribution

The absorption of BC is significantly affected by the particle size (Kahnert, 2010b; Luo et al., 2018b). Therefore, the effects of the size distribution on BC absorption enhancement should be considered carefully. The shape of BC particles is commonly irregular. To describe the size of each BC particle, the radius of the corresponding equivalent volume sphere is typically used. Based on numerous measurements, a lognormal size distribution is observed to fit the realistic BC size distributions well (Bond et al., 2002; Chakrabarty et al., 2006; Wang et al., 2015), and it is widely used in climate models for the estimation of BC radiative forcing (Moffet and Prather, 2009; Chung et al., 2012). However, the mean size and standard deviation vary with the combustion status and aging status. In the atmosphere, geometric mean radii ($r_g$) between $0.05\ um$ and $0.06\ um$ for BC are widely accepted (Alexander et al., 2008; Coz and Leck, 2011b; Liu et al., 2018; Li et al., 2016). The geometric standard deviation ($\sigma_g$) varies within a relatively narrow range. Consequently, bare BC with $r_g$ between $0.03\ um$ and $0.1\ um$ is considered for sensitivity analysis, an $\sigma_g$ from 1.15-1.75. The minimum and maximum equivalent volume radii are $r_{min} = 0.02\ um$ and $r_{max} = 0.2\ um$, respectively.

To estimate the effects of coating thickness on the absorption properties of BC, we assumed that BrC coatings ratio are independent of BC size. The difference between the size distributions of bare BC and coated BC is attributed to the coatings thickness. The size distribution of bare and coated BC is shown in Figure S2. Even though the assumption does not completely agree with the real cases, it is reasonable to make some simplifications for the sensitivity analysis. Here, we must clarify that the size distribution parameters ($r_g$ and $\sigma_g$) mentioned in this work are applied for the bare BC, and the overall effective volume radius of coated BC is equal to the sum of coatings thickness and radius of bare BC.

## 2.6 Calculation of bulk radiative properties of BC

To make our work more consistent with real circumstance, bulk optical properties are considered. These properties are calculated by averaging over a certain particle size distribution. In application, the equivalent volume radii ($r$) of BC are commonly assumed to follow a lognormal size distribution :

$$n(r) = \frac{1}{\sqrt{2\pi}rln(\sigma_g)}exp\left[-\left(\frac{ln(r)-ln(r_g)}{\sqrt{2}ln(\sigma_g)}\right)^2\right] \tag{6}$$

where $r_g$ and $\sigma_g$ represent the geometric mean radius and geometric standard deviation, respectively. Given the size distribution, the bulk $C_{abs}$ can be obtained by the following equation:

$$<C_{abs}> = \int_{r_{min}}^{r_{max}} C_{abs}(r)n(r)dr \tag{7}$$

The bulk $E_{abs}$ and $E_{abs\_internal}$ are calculated as those in equations 3-5. The only difference is that the absorption cross-section is now bulk absorption cross-section.

## 3 Results

### 3.1 Effects of the imaginary part of the BrC refractive index: lensing effect and sunglasses effect

The refractive index of BC is commonly assumed to be wavelength-independent over the visible and near-visible spectral regions, and the imaginary part $k_{BC} \approx 0.79$ (Moosmuller et al., 2009; Bond and Bergstrom, 2006). In addition, Zhang et al. (2017) has demonstrated that the uncertainties of the BC refractive index have little impact on the absorption enhancement of coated BC aggregates. Therefore, a typical refractive index $m = 1.95 + 0.79i$ of BC, was adopted in this study.

The real parts of the BrC refractive indices were assumed to have a constant value of 1.5 (Schnaiter et al., 2005), while the imaginary part of the refractive index ($k_{BrC}$) was significantly dependent on wavelength at shorter visible and ultraviolet (UV) wavelengths (Moosmuller et al., 2009; Andreae and Gelencser, 2006; Alexander et al., 2008). Figure 2 shows the effects of $k_{BrC}$ on $E_{abs}$ and $E_{abs\_internal}$, where $f_{BC}$ represents the BC volume fraction. Large deviations in $E_{abs}$ and $E_{abs\_internal}$ can be observed given different values of $k_{BrC}$. Generally, $E_{abs}$ increases with $k_{BrC}$, while $E_{abs\_internal}$ decreases with increasing $k_{BrC}$. Therefore, it is desirable to evaluate the effects of absorbing coatings on BC absorption enhancement. Given identical $k_{BrC}$ values, the absorption enhancements of thickly-coated BC increase with wavelength. However, for BC that is internally mixed with BrC, wavelength-dependent absorption enhancements are measured to decrease with $\lambda$ (You et al., 2016). This may be due to the wavelength-dependent $k_{BrC}$. For thickly-coated BC, $E_{abs\_internal}$ and $E_{abs}$ decrease with wavelength, but they are not a strong function of $\lambda$ for thinly-coated BC. In addition, compared with BC with non-absorbing coatings, $E_{abs}$ for thinly-coated BC with absorbing coatings seems to be less wavelength-dependent, while $E_{abs}$ for thickly-coated BC with absorbing materials is more sensitive to wavelength.

Many studies have noticed that the lensing effect can greatly enhance the absorption of BC. However, there is also an opposite effect, which is commonly neglected. As shown in Figure 2, as $k_{BrC}$ increases, the value of $E_{abs\_internal}$ of thickly-coated

BC can be below 1. This indicates that the absorption of BC internally mixed with BrC coatings may be less than the sum of the absorption of an external mixture of BrC coatings and BC when $k_{BrC}$ is large. This phenomenon can be explained from physical insights. When the absorption of the coatings is weak, the light can penetrate the coatings into the BC materials, and the absorption of BC is significantly enhanced by the lensing effect. However, as the coating absorption increases, the light is blocked by the outer coatings. Therefore, the light can not fully and deeply penetrate the absorbing coatings on BC. As a result, the total absorption is less than the sum of the absorption of coatings and BC that are calculated separately. Therefore, there is a need to classify the BrC coating effect into lensing effect ($E_{abs\_lensing}$) and sunglasses effect ($E_{Sunglass}$), which represents the absorption enhancements and blocking effects of coatings, respectively.

Liu et al. (2017) defined the lensing effect as the absorption enhanced by addition of non-black carbon. However, from physical point, for BC with BrC coatings, the definition may be not clear as BrC also absorbs solar radiation, and it can be confused with $E_{abs}$. Therefore, here we redefine the lensing effect as the absorption enhanced by addition of non-absorbing materials. In addition, we assume that the lensing effect of BC with absorbing coatings is the same as those with non-absorbing coatings. Accordingly, $E_{abs\_lensing}$ can be caluated using:

$$E_{abs\_lensing} = \frac{C_{abs\_non-absorbing}}{C_{abs\_bare}} \qquad (8)$$

where $C_{abs\_non-absorbing}$ represents the absorption cross-section of BC with non-absorbing coatings. The total $E_{abs}$ should be contributed to the lensing effect, absorption of BrC shell and the sunglass effect. Therefore, $E_{abs}$ can be expressed by:

$$E_{Sunglass} = -\frac{C_{abs\_coated} - C_{abs\_BrC\_shell} - C_{abs\_non-absorbing}}{C_{abs\_bare}} \qquad (9)$$

Combining Equations 3-9, we can obtain $E_{Sunglass}$, and the negative sign represents that the sunglass effect can cause the decrease of total absorption. According to definition of $E_{Sunglass}$, we can easily obtain the relation that the absorption of BC coated with BrC is less than that of an external mixture of BrC and BC when $E_{Sunglass} > E_{abs\_lensing} - 1$.

The sensitivity of $E_{Sunglass}$ to $k_{BrC}$ is shown in Figure 2. For both thinly- and thickly-coated BC, $E_{Sunglass}$ increases with $k_{BrC}$. Fixing $k_{BrC}$, $E_{Sunglass}$ of thinly-coated BC decreases with wavelength. However, for thickly-coated BC, $E_{Sunglass}$ can increase with wavelength at large $k_{BrC}$ (such as $k_{BrC} = 0.16$). For the thinly-coated BC, the blocking of $E_{Sunglass}$ is less than the enhancement of $E_{abs\_lensing}$ (see Figure 5 and Figure 10), therefore, $E_{abs\_internal}$ of thinly-coated BC is larger than 1. For thickly-coated BC, the blocking of $E_{Sunglass}$ can be larger than the enhancement of $E_{abs\_lensing}$ as $k_{BrC}$ is larger, which leads to $E_{abs\_internal}$ of less than 1. The threshold value of $k_{BrC}$ is dependent on particle size and mixing states. Generally, the threshold $k_{BrC}$ value decreases with particle size and coatings thickness, as $E_{abs\_internal}$ of BC thickly-coated with BrC coatings decreases with particle size and coatings thickness in the ultraviolet region (see Figure 6 and Figure 9).

Although core-shell sphere model has been debated for a long time, it is still widely used in climate models. By Combining the electron tomography (ET) and discrete dipole approximation (DDA) method, Adachi et al. (2010) found that the absorption of BC with fluffy structures is significantly enhanced by a core-shell structure at $\lambda = 0.55 \ um$. However, for thickly-coated BC, BC absorption is underestimated at the UV, visible, and IR wavelengths (Kahnert et al., 2012). Mishchenko et al. (2014) has also demonstrated that the $C_{abs}$ of thickly-coated BC with non-absorbing coatings is significantly underestimated by a

core-shell sphere, and investigated the effects of off-center of BC. Their results indicated that the $C_{abs}$ of aged BC covered with thick non-absorbing coatings are approximately 1.44 times higher than those calculated with a core-shell sphere model. Nevertheless, the effects of coating absorption on the applicability of the core-shell sphere model have not been evaluated. As shown in Figure 3, $C_{abs}$ for thinly-coated BC is enhanced by a core-shell sphere structure in the visible spectral region, which agrees with the study of Adachi et al. (2010), while it is underestimated in the ultraviolet region. In addition, the ratio of $C_{abs}$ of thinly-coated BC to core-shell sphere model increases with $k_{BrC}$. However, the applicability of the core-shell sphere model to thickly-coated BC is diverse. Consistent with Kahnert et al. (2012), thickly-coated BC absorption is underestimated by core-shell sphere model when coated with non-absorbing materials. Nevertheless, as $k_{BrC}$ increases, the underestimation becomes insignificant. The reason may be that less light can penetrate deeply into the BC as the $k_{BrC}$ increases, which leads to less variation in absorption. Therefore, the morphological effects of BC are relatively small.

The $E_{abs}$, compared with that for core-shell sphere model, is also calculated. For thinly-coated BC, the $E_{abs}$ is significantly overestimated by core-shell sphere model. However, this overestimation is alleviated by an increasing $k_{BrC}$. For BC that is thickly-coated with non-absorbing materials, the $E_{abs}$ is underestimated by core-shell sphere model at all wavelengths, while it decreases as $k_{BrC}$ becomes larger. The $E_{abs}$ can be overestimated by core-shell sphere model in the ultraviolet spectral region when $k_{BrC}$ is large. Therefore, the absorption characteristics of BC are significantly affected by the absorption of coatings. To agree with the measurements, typical $k_{BrC}$ values are assumed according to Kirchstetter et al. (2004), as shown in Figure S2. In this work, $k_{BrC}$ values of 0.168, 0.114, 0.0354 and 0.001 were assumed for 4 typical wavelengths ($\lambda =350$ $nm$, 404 $nm$ , 532 $nm$ and 700 $nm$, respectively) via interpolation.

### 3.2   Bulk radiative properties: effects of the size distribution

The sensitivity study conducted by Zhang et al. (2017) showed that the $E_{abs}$ for aged BC was significantly affected by the size distribution. They reported different $E_{abs}$ values of $\sim$1.7 - 2.4 and $\sim$ 2.0 - 2.7 for accumulated and coarse modes, respectively. By setting the fractal dimension to be 2.2 and $f_{BC}$ to be 40%, the variations in BC absorption enhancements for different particle size distributions are shown in Figure 4 .Generally, weaker absorption enhancement can be observed by increasing $\lambda$ from ultraviolet region to visible region, which is in agreement with the study of You et al. (2016). By defining the monomers radii, Kahnert (2010a) demonstrated that the absorption cross-section is significantly affected by the particle size, and the cubic fit can greatly describe the relations among equivalent volume radii for freshly emitted BC. However, for the absorption enhancement of thinly-coated BC, the effects of size distribution are not obvious. With variations in $r_g$ and $\sigma_g$, the absorption enhancement changes at ranges of $\sim$1.563 - 1.603, $\sim$1.427 - 1.465, $\sim$1.2440 - 1.275 and $\sim$1.146 - 1.169 at $\lambda = 0.35$ $um$, 0.404 $um$, 0.532 $um$ and 0.7 $um$, respectively. The relative uncertainty in the absorption enhancements caused by the size distribution are 2.56%, 2.66%, 2.81% and 2.01%, respectively. The effects of the size distribution on the absorption enhancement of thinly-coated BC are similar at different wavelengths. Generally, $E_{abs}$ has the largest value when both $r_g$ and $\sigma_g$ are extremely small or extremely large.

The absorption of BrC and BC are considered separately in most cases. To investigate the difference between the absorption of internally mixed BC and the total absorption of BrC and BC (calculated separately), $E_{abs\_internal}$ is also calculated.

$E_{abs\_internal}$ of thinly-coated BC is greater in the visible region due to the insignificant sunglass effects. The senstivity of $E_{abs\_internal}$ is also not obvious to the size distribution. With the size distribution varying, $E_{abs\_internal}$ changes in the range of $\sim 1.055 - 1.099, \sim 1.081 - 1.112, \sim 1.132 - 1.147$ and $\sim 1.140 - 1.165$ for $\lambda = 0.35um, 0.404um, 0.532um$ and $0.7um$, respectively, and the relative uncertainties all are below 2%. In addition, $E_{abs\_lengsing}$ shares a similar dependence on size

distribution as $E_{abs}$ in the visible wavelengths. The reason is that the $E_{abs}$ mainly derives from lensing effects due to the weak absorption of coatings. However, for ultraviolet wavelengths, there is a completely different pattern due to the sunglass effect.

As both the lensing effect and sunglass effect may affect the $E_{abs\_internal}$, $E_{abs\_lensing}$ and $E_{Sunglass}$ are also investigated, and the results are shown in Figure 5. Here $E_{abs\_lensing} - 1$ represents the $E_{abs}$ enhancement caused by the lensing effect, and $E_{Sunglass}$ is the $E_{abs}$ decrease caused by the sunglass effect. For thinly-coated BC, although both $E_{abs\_lensing}$ and $E_{Sunglass}$

decrease with increasing $\lambda$, compared with $E_{Sunglass}$, $E_{abs\_lensing}$ has less spectral-dependence. $E_{abs\_lensing} - 1$ is in the range of $\sim 0.205 - 0.283, \sim 0.186 - 0.251, \sim 0.163 - 0.2$ and $\sim 0.147 - 0.171$ for $\lambda = 0.35\ um, 0.404\ um, 0.532\ um$ and $0.7\ um$, respectively. However, $E_{Sunglss}$ can reach approximately 0.2 at $\lambda = 0.35\ um$, but is about 0 at $\lambda = 0.7\ um$. In addition, for thinly-coated BC, the enhancements of the lensing effect is stronger than the blocking of the sunglass effect. Therefore, $E_{abs\_internal}$ is above 1 for thinly-coated BC (as shown in Figure 2 and Figure 4).

Figure 6 illustrates the effects of size distribution on $E_{abs}$ and $E_{abs\_internal}$ of thickly-coated BC. Compared with thinly-coated BC, there is a different effect pattern for thickly-coated BC. For ultraviolet wavelengths (e.g., $\lambda = 0.35\ um$ and $0.404$ $um$), absorption enhancements decrease as $r_g$ or $\sigma_g$ increases. This indicates that as the particle becomes larger or the size distribution becomes wider, the absorption enhancements become weaker. However, for the visible wavelengths, the effects of the size distribution is quite complicated. The absorption enhancements are relatively small when both $r_g$ and $\sigma_g$ are extremely

large or small. The peak value commonly occurs when $\sigma_g$ is extremely small. Zhang et al. (2017) concluded that the $E_{abs}$ of aged BC is more sensitive to the size distribution in the accumulation mode (where the $\sigma_g$ is relatively small), while the $E_{abs}$ of coarsely coated BC aggregates (i.e., with large $\sigma_g$) show little variation with $r_g$. This is precisely true for BC with weak absorbing coatings, as shown in the results for $\lambda = 0.7\ um$. However, for BC with absorbing coatings, $E_{abs}$ is sensitive to the size distribution for both modes. When fixing the $f_{BC}$ to be 6%, as $r_g$ and $\sigma_g$ vary, the absorption enhancements change in

the ranges of $\sim 3.7 - 7.1, \sim 3.85 - 5.80, \sim 3.06 - 3.74$ and $\sim 1.63 - 2.59$ for $\lambda = 0.35\ um, 0.404\ um, 0.532\ um$ and $0.7\ um$, respectively, and the uncertainties in $E_{abs}$ can reach up to 91.9%, 50.7%, 22.2% and 60.7%, respectively.

$E_{abs\_internal}$ of thickly-coated BC is also significantly affected by the size distribution. With the $r_g$ varying in the range of $0.03-0.1\ um$ and $\sigma_g$ varying in the range of 1.15-1.75, $E_{abs\_internal}$ varies in the range of 0.871-1.053, 0.891-1.121, 1.115-1.383, 1.615-2.442 for $\lambda = 0.35\ um, 0.404\ um, 0.532\ um$ and $0.7\ um$, respectively. In addition, effects of the size distribution

on $E_{abs\_internal}$ and $E_{abs}$ are related to wavelength. $E_{abs\_internal}$ decreases with particle size (i.e, increasing $r_g$) at ultraviolet wavelengths, while it increases as the particles become larger at visible wavelengths (also see Figure S6). Based on physical insights, the reason may be due to two aspects. When the wavelength is in the ultraviolet region, the absorption of the coatings is large, therefore, the blocking effects of the coatings is obvious. Given identical $f_{BC}$ values, the superficial area of the outer coating becomes lager as the particle size increases. As a result, the blocking effects of the outer coatings increases. Therefore,

the $E_{abs\_internal}$ decreases. At visible wavelengths, the absorption of the coatings is negligible, and the light can penetrate

deeply into BC. At that point, the main factor is the enhancement of the lensing effect, and the larger particles may cause a larger superficial area, which leads to the enhanced $E_{abs\_internal}$. $E_{abs}$ shares similar dependences on the size distribution for different wavelengths. In addition, $E_{abs\_internal}$ can be less than 1. This means that the enhancement of the lensing effect is less than the blocking of the sunglass effect. In climate models, the $\sigma_g$ is commonly assumed to be a fixed value , and the BC size distribution is commonly assumed to be in accumulation mode (Liu et al., 2012). The effects of the size distribution at fixed $\sigma_g = 1.5$ are supplemented in Figure S4-S7.

The effects of the size distribution on the lensing effect and sunglass effect of thickly-coated BC are shown in Figure 7. $E_{abs\_lensing}$ is in the range of 2.197-2.514, 2.045-2.486, 1.844-2.526, 1.6147-2.568 at $\lambda = 0.35\ um$, $0.404\ um$, $0.532\ um$ and $0.7\ um$, respectively. It seems that $E_{abs\_lensing}$ is more sensitive to the size distribution in the visible region compared with ultraviolet region. However, $E_{abs\_lensing}$ at different wavelengths does not deviate largely, and the uncertainty is within 25%. However, effects of the size distribution on $E_{Sunglass}$ largely depend on the wavelength. Fixing $f_{BC} = 6\%$, $E_{Sunglass}$ is in the range between 1.586-2.062 at $\lambda = 0.35\ um$, while in the range between 0.001-0.027 at $\lambda = 0.7\ um$. In addition, from Figure 7, we can also see that the enhancement of the lensing effect (represented by $E_{abs\_lensing} - 1$) is less than the blocking of sunglass effect in the ultraviolet region for thickly-coated BC, while the opposite phenomenon is observed in the visible region.

### 3.3 Bulk radiative properties: effects of the composition ratio

To make our calculation meaningful, we compare the calculated $E_{abs}$ and mass absorption cross-section (MAC) with the measurements of Liu et al. (2015b). The measurement results for $E_{abs}$ and MAC are estimated from Figure 1 and supplementary Figure 2 of Liu et al. (2015b). MAC is calculated using:

$$MAC = C_{abs\_coated}/m_{BC} \tag{10}$$

$$m_{BC} = \int_{r_{min}}^{r_{max}} \rho_{BC} \frac{4}{3}\pi r^3 n(r)dr \tag{11}$$

where $m_{BC}$ and $\rho_{BC}$ represent the mass and mass density of BC, respectively. To agree with measurements, the coating thickness is determined by the mass ratio of BrC and BC components $M_R$. In this study, $M_R$ is calculated by:

$$M_R = \rho_{BrC}.(1 - f_{BC})/(\rho_{BC}.f_{BC}) \tag{12}$$

where $\rho_{BrC}$ represents the mass density of BrC. In this work, we assume ambient BC mainly composed of primary organic matter with a low degree of oxidation. Based on the study of Nakao et al. (2013), an OC mass density range of 1-1.2 $g/cm^3$ has been used by Liu et al. (2017). $\rho_{BrC} = 1.1\ g/cm^3$ is assumed in this work. For the BC mass density, the study of Horvath (1993) gives values of $\rho_{BC} = 0.625\ g/cm^3$ and $\rho_{BC} = 1.125\ g/cm^3$. However, Fuller et al. (1999a) pointed out that the values may be not representative for BC in the atmosphere. Medalia and Richards (1972) and Janzen (1980) suggested $\rho_{BC}$ in the range of 1.8-1.9 $g/cm^3$, while Bergstrom (1972) found that the $\rho_{BC}$ value of 1.9-2.1 $g/cm^3$. Bond and Bergstrom (2006)

suggested to use a value of 1.8 $g/cm^3$. Figure S8 compares the computations with measurements by assuming $\rho_{BC} = 1.8$ $g/cm^3$. We assume that $E_{abs}$ and MAC at $\lambda = 0.7$ $um$ do not deviate largely with those at $\lambda = 0.781$ $um$. Modeled $E_{abs}$ at $\lambda = 0.7$ $um$ agrees well with the measurements. Although $E_{abs}$ at $\lambda = 0.404$ $um$ seems to be relatively higher than the measurements, it dose not deviate largely with the measurements. However, modeled MAC is a little smaller than the measured

MAC. Similar results were obtained for bare BC ((Kahnert, 2010b), (Liu and Mishchenko, 2005)) . Therefore, $\rho_{BC} = 1.8$ $g/cm^3$ may be a little high for estimation of MAC.

Bond and Bergstrom (2006) concluded that MAC value of $7.5 \pm 1.2$ $m^2/g$ for bare BC can be assumed at $\lambda = 0.55um$ by reviewing 21 publications of MAC measurements. However, our calculated MAC of 6.02-6.2 $m^2/g$ (see Table 2) at $\lambda = 0.532um$ lies below the range of MAC values suggested by Bond and Bergstrom (2006). Similar conclusions were drawn

by Kahnert (2010b) and Liu and Mishchenko (2005). However, our calculated MAC agrees well with the calculated MAC of $6.0 \pm 0.1$ $m^2/g$ by Kahnert (2010b) at $\lambda = 0.55$ $um$. As MAC depends significantly on BC mass density, to agree with measurements, Liu and Mishchenko (2005) used $\rho_{BC} = 1.0$ $g/cm^3$. However, as pointed by Kahnert (2010b), the measured MAC and modeled MAC were not at the same wavelength, therefore leading to too low retrieved $\rho_{BC}$. To raise the computed MAC values to the average observed value of MAC = $(7.5 \pm 1.2)$ $m^2/g$, $\rho_{BC} = $ 1.3-1.4 $g/cm^3$ was suggested by Kahnert

(2010b). However, this $\rho_{BC}$ value is rather drastic smaller than the value suggested by Bond and Bergstrom (2006). Therefore, Kahnert (2010b) suggested to assume $\rho_{BC} = 1.5 \sim 1.7$ $g/cm^3$ to raise the computational MAC results to the lower bound of the observations. By assuming $\rho_{BC} = 1.5$ $g/cm^3$, the comparison of modeled MAC and $E_{abs}$ with measurements is shown in Figure S9. Overall, the modeled MAC and $E_{abs}$ agree relatively well with the measurement by assuming $\rho_{BC} = 1.5$ $g/cm^3$. Therefore, $\rho_{BC} = 1.5$ $g/cm^3$ is assumed in this study. In addition, according to the previous studies (Liu et al., 2015a; Zhang

et al., 2016), the shell/core ratio $D_p/D_c$ (equivalent particle diameter divided by BC core diameter) was observed to be commonly in the range of $1.1 \sim 2.7$, and the corresponding $M_R$ is approximately $0.24 \sim 13.9$. Therefore, $M_R$ of $0 \sim 13.9$ is considered in this work.

Figure 8 compares the $E_{abs}$ and $E_{abs\_internal}$ for thinly-coated BC with different fractal dimensions at different composition ratios. Following Liu et al. (2018), a $r_g$ of 0.06 $um$ and a $\sigma$ of 1.5 are assumed to reflect the real size distribution of BC. It

is expected that as the coatings increase, $E_{abs}$ becomes much stronger. With $M_R$ varying from 0 to 2.93, $E_{abs}$ variations of $\sim$1 - 2.5, $\sim$1 - 2.2, $\sim$1 - 1.6 and $\sim$1 - 1.285 are obtained for $\lambda = 0.35$ $um$, 0.404 $um$, 0.532 $um$ and 0.7 $um$, respectively. The $E_{abs}$ for thinly-coated BC with weakly-absorbing materials (i.e., $\lambda = 0.7$ $um$) is significantly lower than that for core-shell sphere, as reported by Zhang et al. (2017), where $E_{abs}$ can reach approximately 1.5 when the shell-core ratio is 1.6 ($M_R = 2.2709$) at $\lambda = 0.55$ $um$. Even though the results are gained at two different wavelengths, the $E_{abs}$ for BC that is

coated with weakly absorption coatings should not deviate substantially between $\lambda = 0.55$ $um$ and $\lambda = 0.7$ $um$ (see Figure 12). Therefore, the differences from the previous study are mainly caused by the BC shape, as demonstrated in Figure 3. When the relative contents of BC vary, substantial variations in $E_{abs\_internal}$ can also be observed. As $M_R$ varying in the range of 0-2.93, $E_{abs\_internal}$ increases from 1 to 1.07, 1 to 1.1, 1 to 1.22 and 1 to 1.285 for $\lambda = 0.35$ $um$, 0.404 $um$, 0.532 $um$ and 0.7 $um$, respectively. $E_{abs\_internal}$ of thinly-coated BC increases with $M_R$ in the visible spectral region, while a little decrease in

$E_{abs\_internal}$ can be observed in the ultraviolet region as $M_R$ increases when $M_R$ is larger than a value. This is mainly caused by the blocking of the sunglass effect.

At different wavelengths, the effects of $D_f$ may vary. $E_{abs\_internal}$ increases with $D_f$ in the visible wavelengths, as the more compact structure can lead to a greater lensing interaction. While in the ultraviolet region, as the structure becomes more compact, the interaction of absorbing coatings also increases; therefore, the blocking effects of outer coatings are greater. Therefore, the $E_{abs\_internal}$ can decrease with $D_f$ when $D_f$ is greater than a value. Even though Cheng et al. (2014) and Luo et al. (2018b) showed that the effects of $D_f$ on $C_{abs}$ are not obvious for thinly-coated BC, for $E_{abs}$ of thinly-coated BC, the sensitivity of $D_f$ has not been investigated. To quantify the effects of $D_f$, the relative deviations between $D_f = 1.8$ and $D_f = 2.6$ are also calculated for thinly-coated BC. From Figure 8, we found that the differences in $E_{abs}$ and $E_{abs\_internal}$ among different values of $D_f$ are larger for thicker coatings. Therefore, to evaluate the maximum uncertainty, the $f_{BC}$ is fixed to be 20%. As shown in Figure 9, the differences in $E_{abs}$ and $E_{abs\_internal}$ between $D_f = 1.8$ and $D_f = 2.6$ are all below 5%. $E_{abs}$ of BC thinly coated with non-absorbing coatings is more obviously affected by $D_f$. However, the relative deviations between $D_f = 1.8$ and $D_f = 2.6$ are not exceed 12% (as shown in Figure S10.).

To reveal the factors that comtributes to the complex $E_{abs\_internal}$, the effects of $M_R$ on $E_{abs\_lensing}$ and $E_{Sunglass}$ of thinly-coated BC are investigated at different wavelengths. $E_{abs\_lensing}$ increases with $M_R$ for all wavelengths. It seems that the sensitivity of $E_{abs\_lensing}$ to $M_R$ are more obvious in ultraviolet region compared with visible region. Fixing $D_f = 2.2$, with $M_R$ varying from 0 to 2.93, $E_{abs\_lensing}$ increases from 1 to 1.46, 1.4, 1.32, 1.25 for for $\lambda =$0.35 $um$, 0.404 $um$, 0.532 $um$ and 0.7 $um$, respectively. In addition, more compact structure can result in stronger lensing interaction between monomers, so leads to an $E_{abs\_lensing}$ increase with $D_f$. Moreover, compared with visible region, the effects of $D_f$ are more obvious at ultraviolet region. $E_{Sunglass}$ also increases with $D_f$, as more compact structure may lead to stronger blocking interaction between BC monomers. As expected, $E_{Sunglass}$ is stronger in the ultraviolet region, while tends to be 0 in the visible region. As $M_R$ reaches 2.93, $E_{Sunglass}$ can reach approximately 0.46 at $\lambda = 0.35$ $um$, while $E_{Sunglass}$ is below 0.02 at $\lambda = 0.7$ $um$.

Figure 11 demonstrates the absorption enhancements of thickly-coated BC at different wavelengths for different composition ratios. Similar to thinly-coated BC, $E_{abs}$ increases with increasing $M_R$ or decreasing $\lambda$. When setting $r_g = 0.06$ $um$ and $\sigma_g = 1.5$, as $M_R$ varies from 6.6 to 13.9, $E_{abs}$ increases from 3.4 to 5.4 and 3.25 to 5.2 for $\lambda = 0.35$ $um$ and $\lambda = 0.404$ $um$, respectively, while $E_{abs}$ varies from 2.78 to 3.96 and 2.2 to 2.4 for $\lambda = 0.532$ $um$ and 0.7 $um$, respectively. In addition, the $E_{abs}$ seems to be more sensitive to the composition ratios in the ultraviolet wavelengths. This may be caused by the absorption of coatings, which can substantially enhance the total absorption. In addition, combined of $E_{abs}$ values of thinly-coated and thickly-coated BC, $E_{abs}$ range for BC with BrC coatings is much wider than that for BC with non-absorbing coatings ($E_{abs}$ of $\sim$1 - 2.4) (Zhang et al., 2017, 2018)

At visible wavelengths, the $E_{abs\_internal}$ is greater than 1 due to the small blocking effects of BrC. Defining $r_g$ to be 0.06 $um$ and $\sigma_g$ to be 1.5, as $M_R$ varies from 6.6 to 13.9, $E_{abs\_lensing}$ ranges from $\sim$1.222 - 1.337 and $\sim$2.115 - 2.357 for $\lambda = 0.532$ $um$ and 0.7 $um$, respectively. This indicates that the total absorption of BC and BrC can be substantially enhanced by the lensing effects. However, for ultraviolet wavelengths, the $E_{abs\_internal}$ is less than 1. $E_{abs\_internal}$ is within $\sim$0.913 - 0.924 and $\sim$0.956 - 0.974 for $\lambda = 0.35$ $um$ and $\lambda = 0.404$ $um$, respectively. This demonstrates the absorbing coatings can

significantly block the light into BC. Therefore, the total absorption is less than the sum of BrC absorption and BC absorption. In recent studies, the enhancements of lensing effects has gained increasing attention. However, few studies have investigated the blocking effects of absorbing coatings. As a matter of fact, the blocking effect of absorbing coatings is also a significant factor that affects the total absorption, as the $E_{abs\_internal}$ can be below 1. This indicates that the blocking effects of absorbing coatings may be greater than the enhancements of the lensing effects. Therefore, when BC coated with BrC, we should not only focus on the enhancements of the lensing effects but also carefully consider the blocking effects of the coatings.

There is a different dependence on $M_R$ for $E_{abs\_internal}$ at different wavelengths. $E_{abs\_internal}$ increases with $M_R$ at relative long wavelengths (eg. $\lambda =0.7\ um$), while decreases as the coatings become thicker at relative short wavelengths (0.404 $um$ and 0.532 $um$). This phenomenon can also be explained from physical insights. When the wavelength is short, increased thickness of the coatings may lead to a greater sunglass effect, which weakens the total absorption of the coatings and BC. However, at $\lambda = 0.7\ um$, enhanced $E_{abs\_internal}$ can be obtained by increasing the coatings due to the negligible blocking effects of the coatings. In addition, $E_{abs\_internal}$ increases with wavelength due to the decrease in coating absorption (see Figure 2). $E_{abs\_internal}$ of thickly-coated BC is insensitive to $M_R$ at $\lambda = 0.35$ due to the similar variations of $E_{abs\_lensing}$ and $E_{Sunglass}$ with $M_R$. As $M_R$ varies from 6.6 to 13.9, $E_{abs\_lesnig}$ increases from 2.204 to 2.363, 2.214 to 2.390, 2.216 to 2.473, 2.165 to 2.509 at $\lambda = 0.35\ um$, 0.404 $um$, 0.532 $um$, and 0.7 $um$, respectively. While $E_{Sunglass}$ largely affected by wavelengths. At $\lambda = 0.35\ um$, $E_{Sunglass}$ is in the range from 1.523 to 1.807, while $E_{Sunglass}$ approaches 0 at $\lambda = 0.7\ um$. It is also can be seen from Figure 11 that $E_{Sunglass} > E_{abs\_lensing} - 1$ at $\lambda = 0.35\ um$ and 0.404 $um$. Therefore, $E_{abs\_internal}$ is less than 1.

You et al. (2016) demonstrated that there are different wavelength dependencies for BC that is coated with absorbing and weak absorbing materials. $E_{abs}$ for BC coated with humic acid was observed to vary from 3.0 to approximately 1.6 as $\lambda$ increased from 0.554 $um$ to 0.84 $um$, while it seemed to be essentially wavelength-independent for BC that is coated with sodium chloride. Figure 12 compares the wavelength dependencies of BC coated with non-absorbing materials and BrC. For thinly-coated BC, there are substantial wavelength dependencies for BC coated with BrC. By setting $f_{BC}$ to be 40%, $E_{abs}$ increases from 1.15 to 1.57 with $\lambda$ varying from 0.7 $um$ to 0.35 $um$, which results in approximately 49.6% increase. However, when coated with non-absorbing materials, $E_{abs}$ exhibits small wavelength-dependences. This leads to approximate 8.7% increases as $\lambda$ decreases from 0.7 $um$ to 0.35 $um$. Furthermore, for thickly-coated BC, $E_{abs}$ is significantly wavelength-dependent for BC with BrC coatings. The decrease in $\lambda$ from 0.7 $um$ to 0.35 $um$ would result in approximate 100% increase in $E_{abs}$, while $E_{abs}$ seems to be essentially wavelength-independent for BC with non-absorbing coatings ($E_{abs\_lensing}$); it is approximately 2.4 when $f_{BC} = 6\%$, which is consistent with the value reported by Zhang et al. (2017). The differences of $E_{abs\_lensing}$ of thickly-coated BC between $\lambda = 0.35\ um$ and 0.7 $um$ are below 6.2%. Therefore, the variation in $k_{BrC}$ should be mainly responsible for the significant wavelength dependencies of $E_{abs}$ for BC with BrC coatings when the wavelength is long. For ultraviolet wavelengths ($\lambda$ from 0.35 $um$ to 0.404 $um$), wavelength dependence of $E_{abs}$ is relatively small, as the $E_{abs}$ may increase with wavelength when $k_{BrC}$ is fixed at a large value (see Figure 2), which can reduce the wavelength dependence. Therefore, the contribution of $k_{BrC}$ to the wavelength dependence should be further analyzed in the ultraviolet wavelengths in the future.

In addition, the $E_{abs\_internal}$ of BC coated with BrC is also significantly wavelength-dependent. Fixing $f_{BC} = 40\%$ and 6%, respectively, with $\lambda$ varying form 0.35 $um$ to 0.7 $um$, $E_{abs\_internal}$ increases from 1.05 to 1.18 and from approximately 0.92 to 2.3, respectively. $E_{abs\_lensing} - 1$ and $E_{Sunglass}$ are also compared in Figure 12. $E_{Sunglass}$ decreases significantly with $\lambda$ for both thinly- and thickly-coated BC. For thinly-coated BC, $E_{abs\_lensing} - 1$ is larger than $E_{Sunglass}$ for all wavelengths.

However, $E_{Sunglass}$ can be stronger than $E_{abs\_lensing} - 1$ in ultraviolet region for thickly-coated BC. This indicates that the total absorption of BC and BrC is weakened by internal mixing. Therefore, the sunglass effect should also be noticed for the estimation of aerosol absorption.

## 4 Summary and Discussion

Using MSTM method, the $E_{abs}$ and $E_{abs\_lensing}$ of BC with BrC coatings were investigated at $\lambda = 0.35um, 0.404um, 0.532um$

and $0.7um$, respectively. The main findings of this work are as follows:

1. Generally, $E_{abs}$ increases with $k_{BrC}$ while $E_{abs\_interanl}$ decreases as $k_{BrC}$ becomes larger. For the thinly-coated BC, $E_{abs\_internal}$ is greater that 1 due to the enhancements of the lensing effects. However, for thickly-coated BC, the $E_{abs\_internal}$ can be less than 1. It indicates the total absorption of BrC and BC is less than sum of BrC and BC absorption individually, which is opposite to BC that is coated with weakly-absorbing coatings. This phenomenon may

be caused by the blocking effects of outer coatings. As the absorption of coatings increases, less light can penetrate into BC materials. Therefore, the total absorption of BrC and BC is weakened, resulting in $E_{abs\_internal}$ of less than 1. This effect is named "sunglasses effect" in this study.

2. $C_{abs}$ of thinly-coated BC is underestimated by core-shell sphere model in the ultraviolet region while overestimated in the visible region. In addition, the ratio of $C_{abs}$ of thinly-coated BC to that of core-shell sphere model increases with

$k_{BrC}$. $E_{abs}$ of thinly-coated BC is enhanced by core-shell sphere while the enhancements are alleviated by increasing $k_{BrC}$. There are different dependencies for thickly-coated BC. $C_{abs}$ of thickly-coated BC is underestimated by core-shell sphere model for all wavelengths while the underestimation becomes negligible as $k_{BrC}$ turns very large. $E_{abs}$ of thickly-coated BC with non-absorbing materials is underestimated by core-shell assumption. However, the ratio of $E_{abs}$ of thickly-coated BC to core-shell sphere model decreases with increasing $k_{BrC}$, and $E_{abs}$ is enhanced by core-shell

sphere in the visible region, when the absorption of coatings is large.

3. To make our calculation more consistent with real circumstance, the bulk absorption was calculated and the $k_{BrC}$ is selected by interpolation based on the study of Kirchstetter et al. (2004). For thinly-coated BC, the effects of size distribution on $E_{abs}$ are not obvious. The uncertainties of size distribution result in $E_{abs}$ differences of less than 2.56%, 2.52%, 2.32% and 2.16% for $\lambda = 0.35$ $um$, 0.404 $um$, 0.532 $um$ and 0.7 $um$, respectively. However, $E_{abs}$ of thickly-

coated BC is quite sensitive to the size distribution. $E_{abs}$ differences of approximately 92% can be obtained as $r_g$ and $\sigma_g$ vary for $\lambda = 0.35um$. In addition, different from $E_{abs}$ of $2.2 \sim 2.4$ for thickly-coated BC with weak absorbing coatings, $E_{abs}$ of $3.4 \sim 5.4$ is observed for BC with BrC coatings at $\lambda = 0.35um$ as $M_R$ is in the range of $\sim$6.6-13.9. For thinly

coated BC, $E_{abs}$ of BC with weak absorbing coatings is in the range of approximately $\sim$1 - 1.3 for $\lambda = 0.7\ um$ (i.e. BC with weakly-absorbing coatings) while a wider range of $\sim$1 - 2.5 is obtained for $\lambda = 0.35\ um$. In sumarry, $E_{abs}$ range of BC with BrC coatings is much wider than that of BC with non-absorbing coatings.

4. The suglass effect and lensing effect are compared at different wavelengths. $E_{sunglass}$ is less than $E_{abs\_lensing} - 1$ for thinly-coated BC. This indicates the blocking of the sunglass effect is less than the enhancement of the lensing effect, so the $E_{internal} > 1$ for thinly-coated BC. However, $E_{sunglass}$ can be larger than $E_{abs\_lensing} - 1$ in ultraviolet region for thickly-coated BC, which leads to $E_{internal} < 1$. Therefore, the absorption of BC thickly-coated with BrC can less than an external mixture of BC and BrC. In visible region, $E_{sunglass}$ is less than $E_{abs\_lensing} - 1$ due to the small sunglass effect.

5. $E_{abs}$ of BC with BrC coatings is more wavelength-dependent than those with non-absorbing coatings. For thinly coated BC, $E_{abs}$ of BC with non-absorbing coatings leads to approximately 8.7% increase as $\lambda$ decreases from $0.7\ um$ to $0.35$ $um$ while the difference can reach approximately 50% for BC with BrC coatings. For thickly coated BC, the decrease of $\lambda$ from $0.7\ um$ to $0.35\ um$ would result in approximately 100% increase of $E_{abs}$ for BC with BrC coatings. However, $E_{abs}$ of BC with non-absorbing coatings seems to be to be essentially wavelength-independent. In addition, for thinly coated BC, the effects of $D_f$ are not obvious for $E_{abs}$ and $E_{abs\_lensing}$. The uncertainties of $E_{abs}$ and $E_{abs\_internal}$ caused by $D_f$ all are less than 5%.

In this work, complex morphologies and mixing states are considered. Although current climate models do not simulate any morphological information of aerosols, many laboratory studies has been conducted to investigate the BC morphologies in different mixing states and in different regions. Therefore, our calculations can be applied according to specific mixing states (such as composition ratios) and regions. However, we acknowledge that the understanding of the relation between BC morphology and the composition ratio is still limited. Therefore, further laboratory investigations for the coated BC morphologies should be conducted in the future.

## 5   Acknowledgments

This work was financially supported by the National Key Research and Development Plan (Grant No. 2016YFC0800100 and 2017YFC0805100); National Natural Science Foundation of China (Grant No. 41675024 and U1733126); Fundamental Research Funds for the Central Universities (Grant No. WK2320000035). We particularly thank Dr. D. W. Mackowski and Dr. M. I. Mishchenko for the MSTM code. We also acknowledge the support of supercomputing center of USTC. We particularly thank the constructive suggestions of the three anonymous reviewers.

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

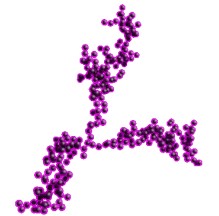 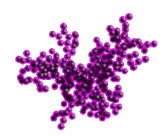 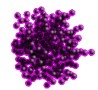 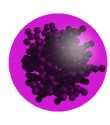

(a) Thinly coated, $D_f = 1.8$     (b) Thinly coated, $D_f = 2.2$     (c) Thinly coated, $D_f = 2.6$     (d) Thickly coated, $D_f = 2.6$

**Figure 1.** Typical morphologies of BC, $n_s = 300, k_0 = 1.2$.

**Table 1.** Morphological parameters of BC aerosols

| Parameters | Thinly-coated BC | Thickly-coated BC |
|:---:|:---:|:---:|
| $k_o$ | 1.2 | 1.2 |
| $n_s$ | 1-1000 | 1-1000 |
| $D_f$ | 1.8,2.2,2.6 | 2.6 |
| $f_{soot}$ | 0.2,0.4,0.6,0.8,1.0 | 0.05, 0.06, 0.075, 0.1 |

**Table 2.** MAC $(m^2/g)$ for bare BC at different $D_f$ ($r_g = 0.06\ um, \sigma_g = 1.5$).

| $\lambda\ (nm)$ | $D_f = 1.8$ | $D_f = 2.2$ | $D_f = 2.6$ |
|:---:|:---:|:---:|:---:|
| 350 | 9.30 | 9.03 | 8.48 |
| 404 | 8.14 | 7.95 | 7.60 |
| 532 | 6.20 | 6.11 | 6.02 |
| 700 | 4.68 | 4.64 | 4.65 |

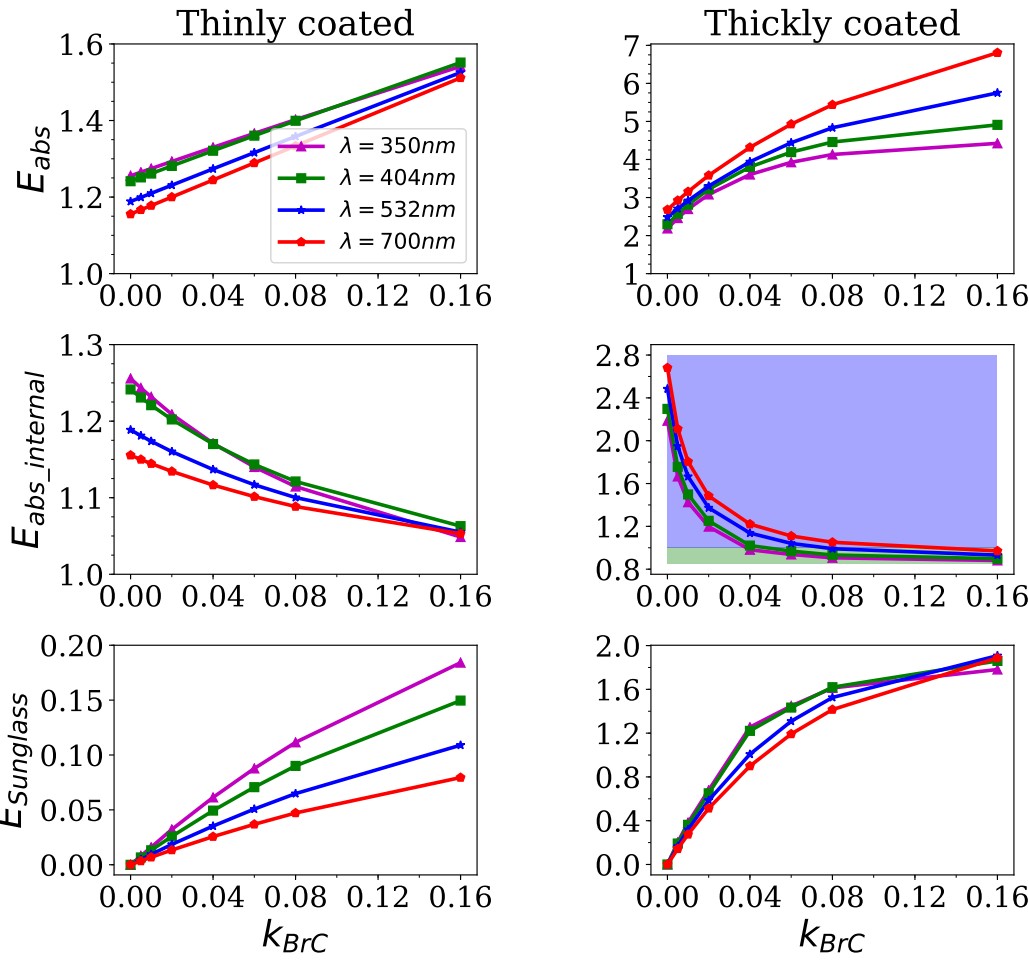

**Figure 2.** Effects of $K_{BrC}$ on specific enhancement ($n_s = 200$). For thinly-coated BC, $D_f = 2.2$, and $f_{BC} = 40\%$; for thickly-coated BC, $D_f = 2.6$, and $f_{BC} = 5\%$. The blue shading represents the $E_{abs\_internal}$ of larger than 1, while the green shading describe the range of $E_{abs\_internal}$ of less than 1.

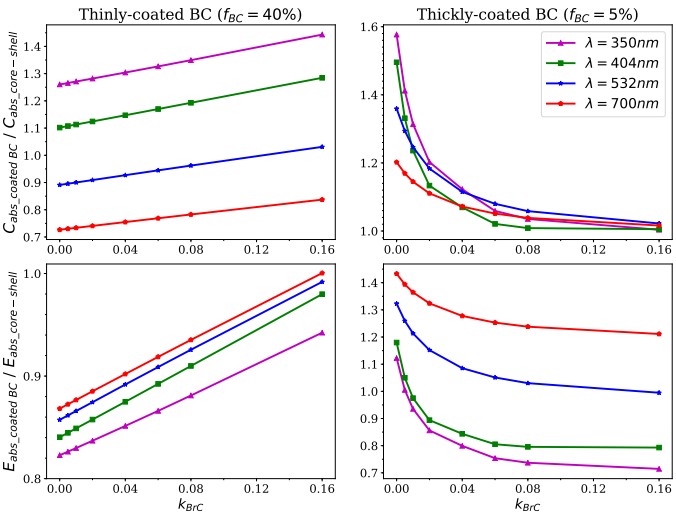

**Figure 3.** Effects of $k_{BrC}$ on the applicability of core-shell sphere ($n_s = 200$).

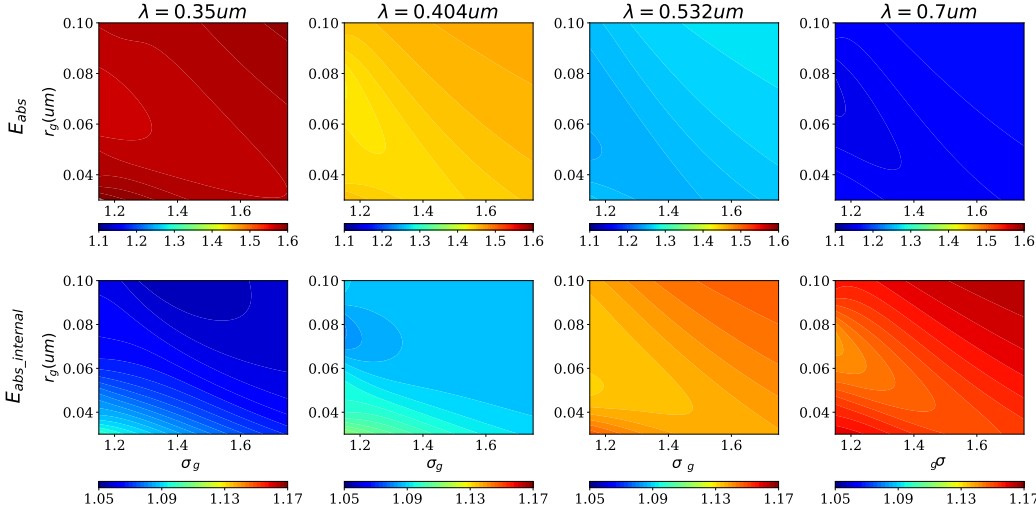

**Figure 4.** $E_{abs}$ and $E_{abs\_internal}$ of thinly-coated BC with BrC coatings at different size distributions ($D_f = 2.2, f_{BC} = 40\%$).

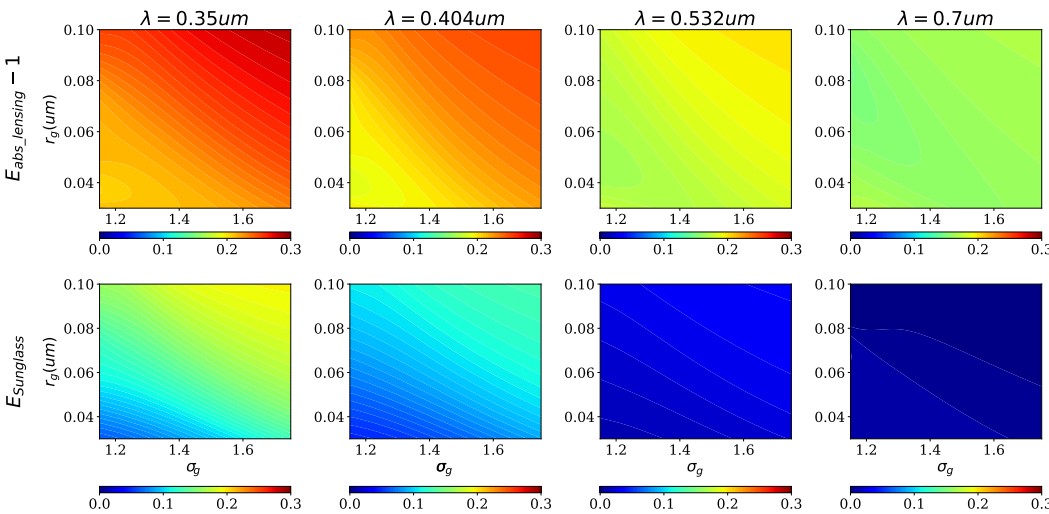

**Figure 5.** Similar as Figure 4, but for $E_{abs\_lensing}$ and $E_{Sunglass}$.

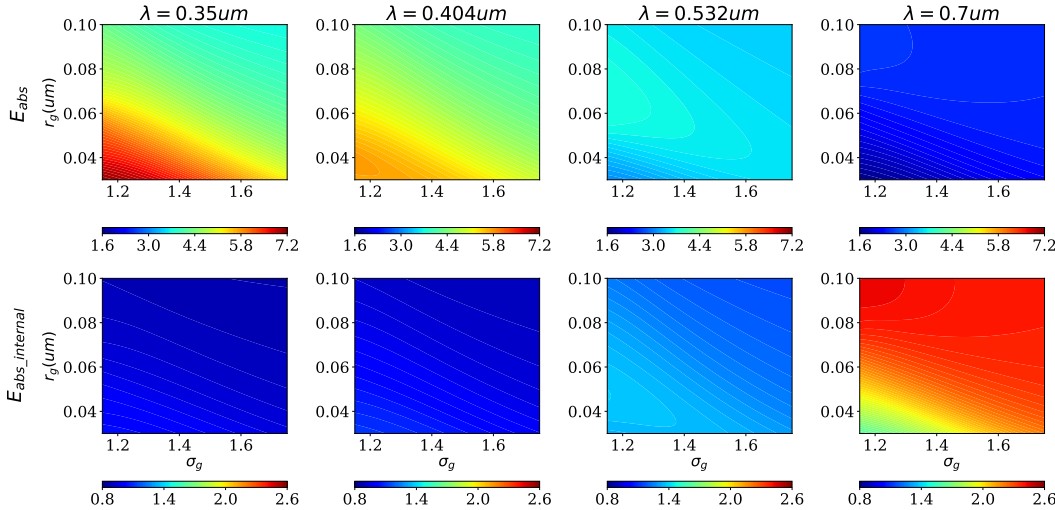

**Figure 6.** $E_{abs}$ and $E_{abs\_internal}$ of BC thickly-coated with BrC at different size distributions ($D_f = 2.6$, $f_{BC} = 6\%$).

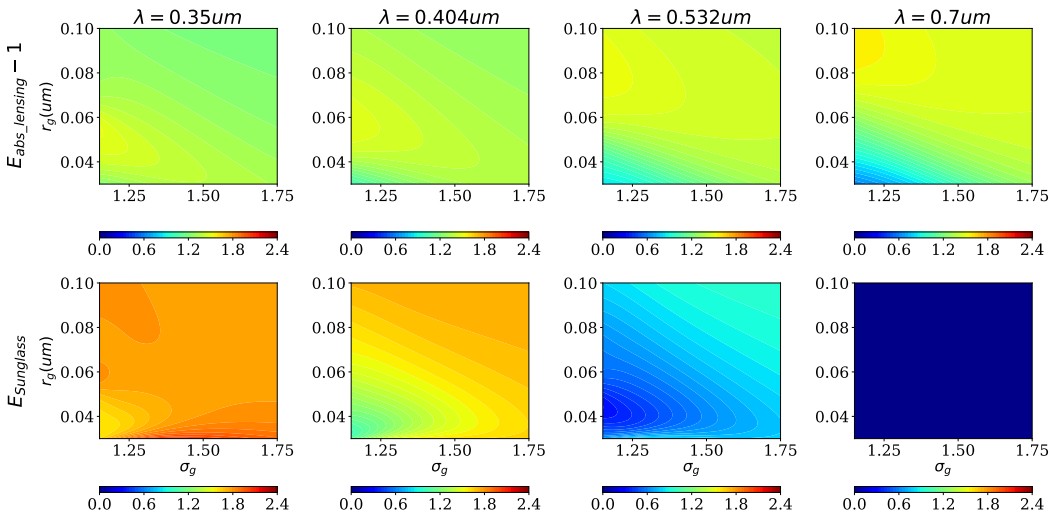

**Figure 7.** Similar as Figure 6, but for $E_{abs\_lensing}$ and $E_{Sunglass}$.

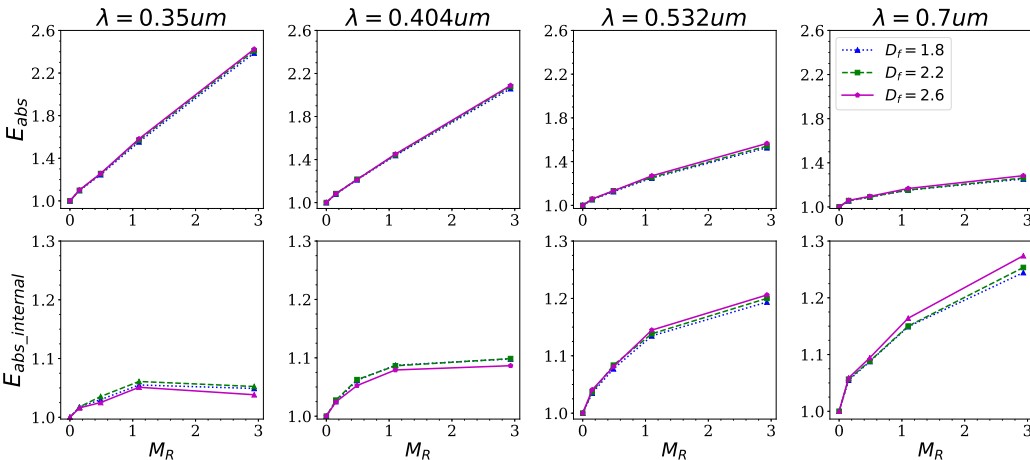

**Figure 8.** $E_{abs}$ and $E_{abs\_internal}$ of thinly-coated BC with BrC coatings varying with $M_R$ for different $D_f (r_g = 0.06um, \sigma_g = 1.5)$.

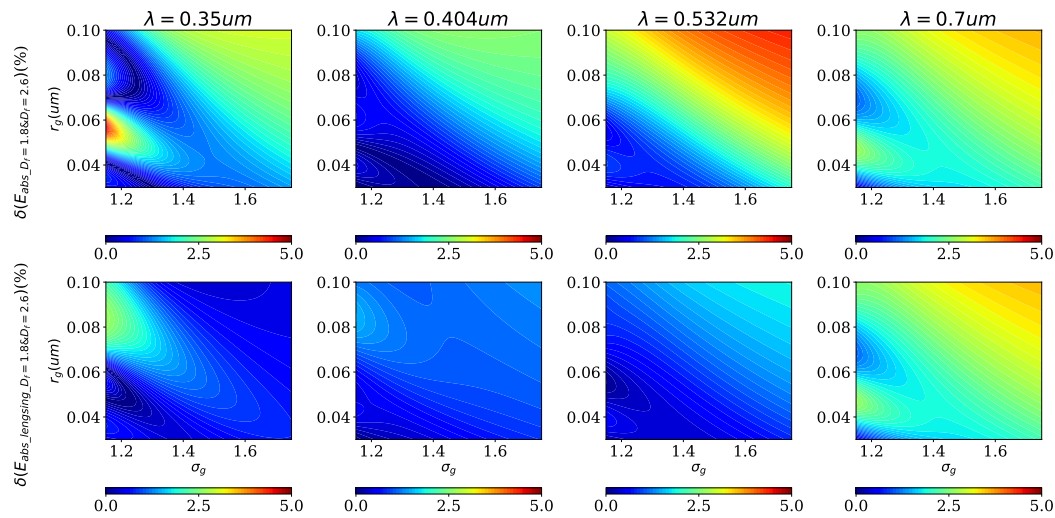

**Figure 9.** The relative deviations of absorption properties between $D_f = 1.8$ and $D_f = 2.6$ for thinly-coated BC with BrC coating ($f_{BC} = 20\%$).

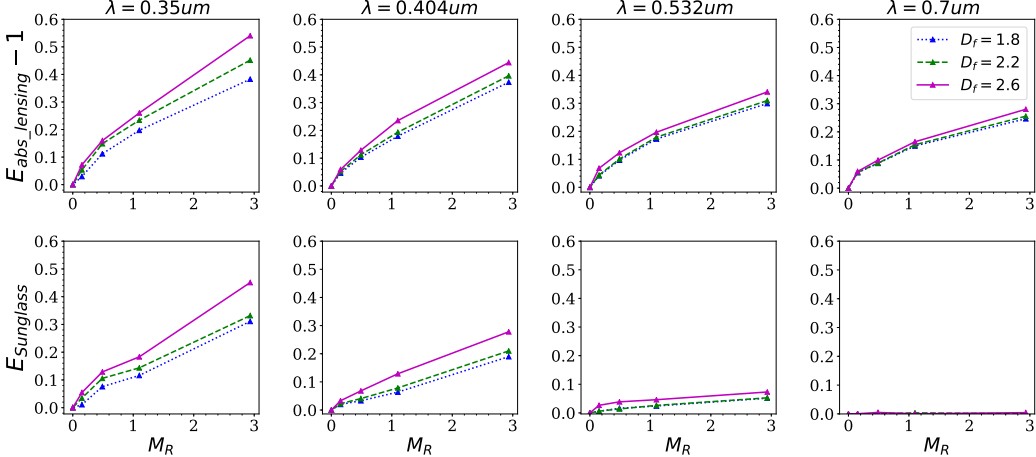

**Figure 10.** Similar as Figure 8, but for $E_{abs\_lensing}$ and $E_{Sunglass}$.

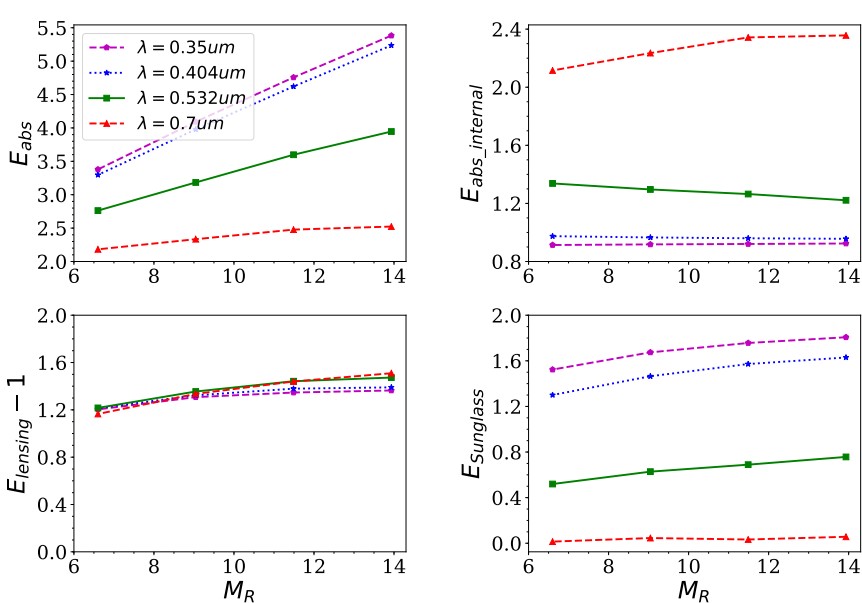

**Figure 11.** $E_{abs}$ and $E_{abs\_lensing}$ of thickly-coated BC with BrC coatings varying with $M_R$ ($r_g = 0.06um, \sigma_g = 1.5$).

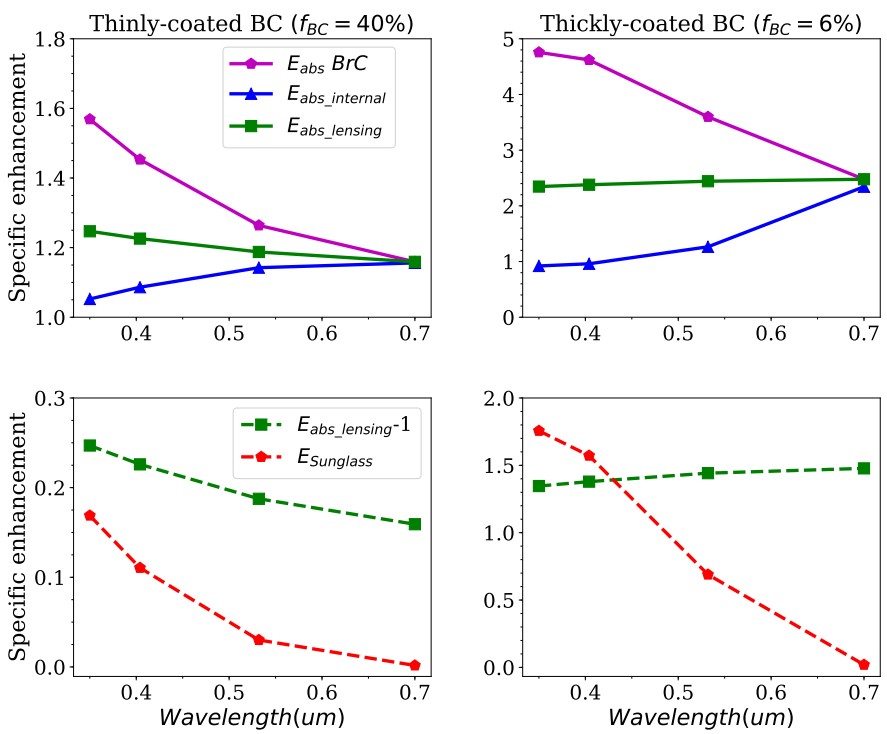

**Figure 12.** Comparison of BC coated with non-absorbing materials and that coated with BrC ($r_g$=0.06$um$, $\sigma_g = 1.5$). $D_f = 2.2$ and $D_f = 2.6$ were assumed for thinly-coated and thickly-coated BC, respectively.