# Peer review of "Effects of brown coatings on the absorption enhancement of black carbon: a numerical investigation"

_Atmospheric Chemistry and Physics, 2018_

## Referee Comment (RC1) · Anonymous Referee #1 · 2 Jul 2018

This is a solid contribution on an important subject. I appreciate the authors' use of state-of-the-art modeling techniques for the study of soot-containing aerosols with highly complex morphologies. The technical content of the paper appears to be correct, and the conclusions are well justified. I have only three minor comments.

1. In Section 2.3, an appropriate generic reference for the DDA would be J. Quant. Spectrosc. Radiat. Transfer 106, 558-589 (2007).

2. The authors model randomly oriented nonspherical aerosols. The use of the model of randomly oriented particles has two aspects (see the recent rigorous analysis in Optics Letters 42, 494-497 (2017)). First, the orientation distribution function mush have

a specific mathematical form, so I wonder whether this is the case with the computer program used to calculate light scattering. Second, technically speaking, the computation for a nonspherical particle must be supplemented by the computation for its mirror image. I wonder whether this was done, or it was found that the two computations yield very close results. These two issues need to be clarified.

3. The authors' analysis of the differences between the effects of absorbing and nonabsorbing shells is quite interesting. It would be instructive to compare their observations with those in Optics Letters 39, 2607-2610 (2014).

---

## Referee Comment (RC2) · Anonymous Referee #2 · 3 Aug 2018

Black carbon (BC) particles mixing with brown carbon (BrC) coatings are simulated by two morphologies, including: thinly and thickly coated states. Light absorption properties of BC-containing particles are calculated using the superposition T-matrix method. The sensitivity of the imaginary part of BrC refractive index on the light absorption is investigated for the realizations with the aerosol ensembles. The authors showed some interesting results, but the effects they presented are not clear and the simulations are not validated by the measurements. While you revise the paper, please take the following into consideration.

1. In this study, the absorption enhancement, lensing effect, blocking effect, sunglasses

effect, and strengthening effect are discussed, but they are confused. In the previous studies, e.g. Liu (2017), "lensing effect is that the addition of non-black-carbon materials to black-carbon particles may enhance the particles' light absorption by 50 to 60% by refracting and reflecting light". The clear definitions of these effects are important, because brown coating also absorbs solar radiation itself. In Equation (3), the effect of brown coating on absorption enhancement is not considered, and may generate an unreasonable Eabs value, such as 5.4 (Line 5 in abstract). It would more appropriate to compare the absorption of BC coated by BrC with an external mixture of BrC and BC, rather than bare BC alone. In Equation (4), how to calculate far-field results of Cabs_brc (total_size) and Cabs_brc (bare_size) in the BC-BrC mixtures, and do you considered the complex morphologies of BrC in 'total size' cases? In bare BC, the BrC coating may be not exist. The 'lensing effect' is widely used in the climate studies, thus, Equation (5) may be potentially misleading. It is necessary to clearly explain these effects.

Referece: Liu, Dantong, et al. "Black-carbon absorption enhancement in the atmosphere determined by particle mixing state." Nature Geoscience 10.3 (2017): 184.

2. Line25-27, Page 2. "Nevertheless, in the atmosphere, there is a type of organic carbon that absorbs the radiation in the range of the ultraviolet and visible spectra, which is known well as brown carbon (BrC); BC can also be mixed with BrC." Please give the references about the morphologies and mixing states of BC-BrC mixtures to support this simulation.

3. In Section 3.2 Bulk radiative properties, it is suggested to estimate the absorption enhancements of BC aerosols by the mass absorption cross section (MAC) rather than the cross section (Cabs), because of the normalization of BC mass. Moreover, the simulations of MAC can be validated by the previous measurements.

4. In the abstract, please define the 'Cabs', 'KBrC' before use them.

5. In Figure 6, 7 and 11, the range of color bar is suggested to be unified.

---

## Referee Comment (RC3) · Anonymous Referee #3 · 4 Aug 2018

**Review Comments for "Effects of brown coatings on the absorption enhancement of black carbon: a numerical investigation" by Luo et al.**

The topic of black carbon (BC) absorption enhancement has been investigated by numerous previous modeling/lab/field studies. The present manuscript systematically quantified the effects of brown carbon coating and associated morphological properties on BC absorption enhancement and proposed a "sunglasses effect", which provides some new understanding in this topic. This study is suitable for ACP and its structure is clear. Before it can be considered for publication, I have a few comments and suggestions to help improve the manuscript.

1. Abstract: The authors mentioned "thickly-coated" and "thinly-coated" here. How thick is "thickly-coated"? Please quantitatively define it here and in the main text as well.

2. Abstract (Lines 11-12): "the uncertainties … have differences of less than 2.6% and 6% …". The expression "uncertainties have differences …" is weird. Please rephrase this sentence.

3. Page 1, Line 20: "second contribution" should be "second contributor". The reference for this sentence should also include Bond et al., 2013 (JGR).

4. Introduction: The authors mentioned that BC absorption enhancement varies significantly in different field measurements, which could be due to complex morphology and mixing state during BC aging processes. But one missing part is the evidence for the complex BC morphology and mixing state observed in field measurements (e.g., *Y. Wang et al.* 2017, doi:10.1021/acs.estlett.7b00418; *S. China et al.* 2013, doi:10.1038/ncomms3122). I suggest including several sentences in the introduction to point out this aspect.

5. Page 4, Lines 8-10: Using Df values to define "thickly-coated" and "thinly-coated" BC is not straightforward. Why not use the coating thickness or mass directly? There may be some situations where Df is smaller than 2.6, but the coating is still more than that of BC with a relatively higher Df.

6. Page 5, Lines 6-10: Recently, another important and efficient particle light-scattering method, the geometric-optics surface-wave (GOS) method (*Liou et al.* 2011, doi.org/10.1016/j.jqsrt.2011.03.007; *C. He et al.* 2016, doi.org/10.1016/j.jqsrt.2016.08.004), has also been developed and applied to resolve complex BC coating structures and showed consistent results with MSTM, which could be included here.

7. Page 6, Lines 13-14: The authors assumed BrC coatings are uniformly distributed over the BC surface, but they also argued that the blocking effect of coating is important, which could be affected by how coating materials are distributed over BC particle surface. Thus, assuming the uniform distribution of BrC coating may lead to nontrivial biases in calculations. Could the authors comment or add some discussions on this?

8. Page 7, Lines 11-12: "Generally, Eabs increases … with increasing kBrC." Is this true for all wavelengths? Please clarify here.

9. Page 7, Lines 16-17: "… compared with BC with non-absorbing coatings, Eabs for thinly-coated BC with absorbing coatings seems to be less wavelength-dependent, …" This is interesting but a little counter-intuitive. Could the authors provide some explanations?

10. Section 3: The authors highlighted two important but opposite effects: conventional lensing effect and sunglasses effect. It is interesting to see how these two effects vary with kBrC, Df, and wavelength. Since the authors already calculated the absorption due to these two effects, it is straightforward to calculate the contributions of these two effects to the total absorption enhancement. This would be very informative and worth discussing. Also, according to the authors' arguments, there should be one critical point (or critical kBrC value) for the two effects to be the same. It would be very interesting to see what this point/value is.

11. Page 8, Line 19: "shorter wavelength". Please give a more quantitative wavelength range.

12. Page 8, Line 15 (and elsewhere): "relative errors". I suggest using "relative uncertainty" instead of "error".

13. Page 12, Line 1: "combined of Eabs …". Should it be "combining Eabs …"?

14. Section 4: Could the authors add some discussions on how to apply their results in this study to climate models? Current climate models do not simulate any morphological information of aerosols and generally assume a core-shell structure or external mixing for aerosols.

---

## Author Comment (AC1) · 4 Oct 2018

**Responses to the comments of the reviewer #1**

(The responses are highlighted in blue)

First of all, we would like to thank the three anonymous reviewers for their thoughtful review and valuable comments to the manuscript. In the revision, we have accommodated all the suggested changes into consideration and revised the manuscript accordingly. All changes are highlighted in the revised manuscript in **BLUE** in the revision.

In this response, the questions and comments of reviewers are in black font, and responses are highlighted in **BLUE**. The changes made in the revised manuscript are marked in RED font.

This is a solid contribution on an important subject. I appreciate the authors' us of state-of-the-art modeling techniques for the study of soot-containing aerosols with highly complex morphologies. The technical content of the paper appears to be correct, and the conclusions are well justified. I have only three minor comments.

1.    In Section 2.3, an appropriate generic reference for the DDA would be J. Quant. Spectrosc. Radiat. Transfer 106, 558-589 (2007).

**Response:** Thanks for pointing it out. We have modified it in the revised manuscript.

2.    The authors model randomly oriented non-spherical aerosols. The use of the model of randomly oriented particles has two aspects (see the recent rigorous analysis in Optics Letters 42, 494-497 (2017)). First, the orientation distribution function mush have a specific mathematical form, so I wonder whether this is the case with the computer program used to calculate light scattering. Second, technically speaking, the computation for a non-spherical particle must be supplemented by the computation for its mirror image. I wonder whether this was done, or it was found that the two computations yield very close results. These two issues need to be clarified.

**Response:** Thanks for your comments. First of all, we agree that the calculations for randomly oriented non-spherical aerosols should be clarified. The non-spherical particles that we considered in this manuscript are atmospheric aerosols. It is reasonable to assume the orientations of black carbon (BC) particles in the atmosphere is completely random, that is to say that the probability of every orientation is identical. As a result, the normalized probability density function of particle directions is nearly a constant. Therefore, Eqs. (3) in the study of Mishchenko and Yurkin, (2017) is satisfied. We have added some text and related reference in the revised manuscript:

"In this study, all the radiative properties of BC were calculated based on the assumption that BC

particles and their mirror counterparts are present in equal numbers in ensemble of randomly oriented particles. In the atmosphere, it is reasonable to assume that the possibility of each particle direction is identical, which mathematically satisfies the definition of random orientation (Mishchenko and Yurkin, 2017)."

Rigorously, the computation for a non-spherical particle should indeed be supplemented by the computation for its mirror image. However, Kahnert (2017) has demonstrated that the calculations for closed-cell model calculated using DDA by numerically averaging over each particle direction and those using MSTM don't deviate largely. This indirectly verifies that the results for randomly oriented non-spherical aerosols are close to their mirror counterparts. In addition, we found little changes by altering the option of target_euler_angles_deg in MSTM. Therefore, we didn't provide the computation for their mirror counterparts.

3.    The authors' analysis of the differences between the effects of absorbing and non-absorbing shells is quite interesting. It would be instructive to compare their observations with those in Optics Letters 39, 2607-2610 (2014).

Response: Thanks for your suggestion. We have compare the results in present study with the results presented in Mishchenko et al. (2014) in the section 3.1 of revised manuscript:

"For aged BC with thick coatings, BC absorption is underestimated at the UV, visible, and IR wavelengths (Kahnert et al., 2012). Mishchenko et al. (2014) has also demonstrated that the $C_{abs}$ of thickly coated with non-absorbing coatings is significantly underestimated by a core-shell sphere, and investigated the effects of off-center of BC. Their results indicated that the $C_{abs}$ of aged BC covered with thickly non-absorbing coatings are approximately 1.44 times higher than those calculated with a core-shell sphere model. Nevertheless, the effects of coating absorption on the applicability of the core-shell sphere model have not been evaluated".

Kahnert, M.: Optical properties of black carbon aerosols encapsulated in a shell of sulfate: comparison of the closed cell model with a coated aggregate model, Opt Express, 25, 24579-24593, 2017.

Mishchenko, M. I., and Yurkin, M. A.: On the concept of random orientation in far-field electromagnetic scattering by nonspherical particles, Opt Lett, 42, 494-497, 2017.

Mishchenko, M. I., Liu, L., Cairns, B., and Mackowski, D. W.: Optics of water cloud droplets mixed with black-carbon aerosols, Opt Lett, 39, 2607-2610, 2014.

---

## Author Comment (AC2) · 5 Oct 2018

**Response to the comments of the Reviewer #2**

(The responses are highlighted in blue)

First of all, we would like to thank the three anonymous reviewers for their thoughtful review and valuable comments to the manuscript. In the revision, we have accommodated all the suggested changes into consideration and revised the manuscript accordingly. All changes are highlighted in the revised manuscript in **BLUE** in the revision.

In this response, the questions and comments of reviewers are in black font, and responses are highlighted in **BLUE**. The changes made in the revised manuscript are marked in RED font.

Black carbon (BC) particles mixing with brown carbon (BrC) coatings are simulated by two morphologies, including: thinly and thickly coated states. Light absorption properties of BC-containing particles are calculated using the superposition T-matrix method. The sensitivity of the imaginary part of BrC refractive index on the light absorption is investigated for the realizations with the aerosol ensembles. The authors showed some interesting results, but the effects they presented are not clear and the simulations are not validated by the measurements. While you revise the paper, please take the following into consideration.

Response: Thanks for your valuable comments. Please see the point-to-point response below to the concerns raised for this manuscript.

1. In this study, the absorption enhancement, lensing effect, blocking effect, sunglasses effect, and strengthening effect are discussed, but they are confused. In the previous studies, e.g. Liu (2017), "lensing effect is that the addition of non-black-carbon materials to black-carbon particles may enhance the particles' light absorption by 50 to 60% by refracting and reflecting light". The clear definitions of these effects are important, because brown coating also absorbs solar radiation itself. In Equation (3), the effect of brown coating on absorption enhancement is not considered, and may generate an unreasonable $E_{abs}$ value, such as 5.4 (Line 5 in abstract). It would more appropriate to compare the absorption of BC coated by BrC with an external mixture of BrC and BC, rather than bare BC alone. In Equation (4), how to calculate far-field results of $C_{abs\_BrC \text{ (total\_size)}}$ and $C_{abs\_BrC \text{ (bare\_size)}}$ in the BC-BrC mixtures, and do you considered the complex morphologies of BrC in 'total size' cases? In bare BC, the BrC coating may be not exist. The 'lensing effect' is widely used in the climate studies, thus, Equation (5) may be potentially misleading. It is necessary to clearly explain these

effects.

Referece: Liu, Dantong, et al. "Black-carbon absorption enhancement in the atmosphere determined by particle mixing state." Nature Geoscience 10.3 (2017): 184.

**Response:** Thanks for your comments and valuable suggestions. We agree that the definition of $E_{abs\_lensing}$ in the previous version of the manuscript is not clear. The sunglass effect, BC absorption enhancement, BrC enhancement, and the lensing effects should be clearly explained.

Usually, "lensing effect is widely used in the climate studies", and they contribute all absorption enhancement of internally mixed BC to the "lensing effect" of coatings. From a physics point of view, we think lensing effects is suitable for non-absorbing coatings, the definition is as follows:

$$E_{len\sin g} = \frac{C_{abs\_coated\_non\text{-}absorbing}}{C_{abs\_bare}}$$

The lensing effect defined in Equation (5) of previous version is the comparison of the absorption of BC coated by BrC with an external mixture of BrC and BC. It is resulted from the interaction of lensing effect and sunglass effect we defined.

Liu et al. (2017a) defined the lensing effect as the absorption enhanced by addition of non-black carbon. However, from the physical point of view, for BC with BrC coatings, the definition may be not clear, and it can be confused with $E_{abs}$. Therefore, we redefined the lensing effect as the absorption enhanced by addition of non-absorbing coatings in the revised manuscript. In addition, we assume that the lensing effect of BC with absorbing coatings is the same as those with non-absorbing coatings. We believe this is a reasonable assumption since the BrC and nonabsorbing coating have a similar value of real part of refractive index.

We agree that it is a valuable suggestion to compare the absorption of BC coated by BrC with an external mixture of BC and BrC. Actually, the $E_{abs\_lensing}$ defined in the previous version of the manuscript has compared the absorption of BC coated by BrC with an external mixture of BC and BrC. However, the comparison is not in a direct manner. Therefore, we define a new parameter, $E_{abs\_internal}$, to represent the ratio between the absorption of BC coated by BrC and an external mixture of BC and BrC.

However, in the revised manuscript, the $E_{abs}$ was compared with measurement results in the literatures, while the $E_{abs\_internal}$ was not. It is because the $E_{abs}$ were commonly measured while usually $E_{abs\_internal}$ results were not available by measurements.

The absorption of BrC shell is calculated by the absorption of BrC that is with the same shape as the coated BC subtracting the absorption of BrC that is with the same shape as the bare BC, as shown in Equation (4) of previous version. The calculation of BrC shell is illustrated in Figure 1 in this response (Figure S1 in the revised manuscript). In this process, we assume that the absorption of BrC with the same shape as the coated BC is identical as the external mixture of BrC with the same shape as bare BC and BrC shell. We must clarify that this process neglects the blocking effect and lensing effect of outer BrC shell on the internal BrC. However, as the BrC absorption is significantly less than the BC absorption with identical shape, the absorption caused by the blocking effect and lensing effect of outer BrC on the internal BrC is relative small compared with the BC absorption. Therefore, it is reasonable to make some simplifications.

[Figure]

Figure 1 Calculation of the absorption of BrC shell.

Therefore, we indeed considered the complex morphologies of BrC in 'total size' cases. We think "$C_{abs\_BrC\_(total\_size)}$" and "$C_{abs\_BrC\_(bare\_size)}$" may be a little confusing. For this reason, we have changed "$C_{abs\_BrC\_(total\_size)}$" and "$C_{abs\_BrC\_(bare\_size)}$" into "$C_{abs\_BrC\_(coated\ shape)}$" and "$C_{abs\_BrC\_(bare\ shape)}$". It is true that the BrC coating don't exist in the case of bare BC. However, Equation (4) is aimed to calculate the BrC shell of absorption.

We clearly defined the sunglass effect in the revised manuscript. We contribute $E_{abs}$ of BC with BrC coatings to lensing effect, BrC absorption enhancement and sunglass effect. Therefore:

$$E_{Sunglass} = -\frac{C_{abs\_coated} - C_{abs\_BrC\_shell} - C_{abs\_non-absorbing}}{C_{abs\_bare}}$$

The negative sign represents that the sunglass effect can cause the decrease of total absorption. Combining all the definitions, we can easily obtain the relation that the absorption of BC coated with BrC is less than that of an external mixture of BrC and BC when $E_{Sunglass} > E_{abs\_lensing} - 1$.

One may confused about the calculation of coated BC with the BrC shell. Coated BC was calculated directly using MSTM. The calculation of the absorption caused by the Sunglass effect is shown in Figure 2 of this response.

Figure 2 Calculation of the absorption caused by the sunglass effect. $C_{abs\_Sunglass}$ represents the absorption cross-section caused by the sunglass effect.

2. Line25-27, Page 2. "Nevertheless, in the atmosphere, there is a type of organic carbon that absorbs the radiation in the range of the ultraviolet and visible spectra, which is known well as brown carbon (BrC); BC can also be mixed with BrC." Please give the references about the morphologies and mixing states of BC-BrC mixtures to support this simulation.

**Response:** Thanks for your comments. As the BC ages in the atmosphere, BC becomes more compact and other materials (such as sulfate and organic substances) can condense onto the particles. The organic coating can be POA or SOA. BC can be embedded in a non-BC shell (Wang et al., 2017) (China et al., 2013). When non-BC fraction is low, BC can still present fractal structure, as demonstrated in the Figure 1 (a-3) of Wang et al. (2017) (Figure 3a in this response). As BC is further coated, BC becomes more compact and the coating shell becomes more spherical (Lewis et al., 2009), as shown in the Figure 3 of China et al. (2013) (Figure 3b in this response).

Eventually, BC aggregates are collapsed into more compact and spherical clusters when fully engulfed in coating material (referred as thickly-coated BC in this study) (Zhang et al., 2008b). We added the references about the morphologies and mixing states of BC in the introduction of revised manuscript:

"Freshly emitted BC commonly presents fractal structures. As the BC ages in the atmosphere, BC becomes more compact and OC materials can condense onto the particles. Therefore, BC can be embedded in an OC shell (China et al., 2013a; Wang et al., 2017). When non-BC fraction is low, BC can still present near fractal structure (referred as thinly coated in this study) (Wang et al., 2017). As BC is further coated, BC aggregates are collapsed into more compact and spherical clusters when fully engulfed in coating material (referred as thickly-coated BC in this study) (Zhang et al., 2008b)."

[Figure]

(a)                                              (b)

Figure 3 Typical morphologies of coated BC. (a) Thinly-coated BC with fractal structure (Wang et al., 2017); (b) Coated BC with more spherical coating (China et al., 2013).

3. In Section 3.2 Bulk radiative properties, it is suggested to estimate the absorption enhancements of BC aerosols by the mass absorption cross section (MAC) rather than the cross section (Cabs), because of the normalization of BC mass. Moreover, the simulations of MAC can be validated by the previous measurements.

**Response:** Thanks for your valuable suggestions. The calculation of $E_{abs}$ was not estimated by mass absorption cross section (MAC), the reason is that the absorption enhancement is defined as the amplification of total absorption but not the amplification of mass absorption. Although many studies used the MAC to estimate $E_{abs}$, MAC is calculated as the mass absorption per unit mass of BC but not per unit mass of total BC-containing aerosols. Therefore, the MAC can be calculated using:

$$MAC_{coated/bare} = C_{abs\_coated/bare} / m_{BC}$$

Where $m_{BC}$ represents the mass of BC.

The $E_{abs}$ can be expressed as:

$$E_{abs} = \frac{MAC\_abs\_coated}{MAC\_abs\_bare} = \frac{C_{abs\_coated}}{C_{abs\_bare}}$$

This is consistent with the definition of $E_{abs}$ in our work.

We do agree that our calculations should be validated by measured results. We compared the modeled $E_{abs}$ and MAC with the results of Liu et al. (Liu et al., 2015), and the comparison is supplemented in support information, as shown in the Figure S1. The measured $E_{abs}$ values were estimated from Figure 1 of Liu et al. (Liu et al., 2015). In this work, we assume the $E_{abs}$ and MAC at λ=700 nm and 781 nm do not deviate largely. The comparison of measured and modeled MAC is shown in Figure S7 and Figure S8 in the revised manuscript. When BC mass density is assumed to be 1.5g/cm³, the calculated MAC and $E_{abs}$ are relatively well in agreement with the measurements (see Figure 5 in this response or Figures S8-S9 in the revised manuscript). As for why assuming BC mass density to be 1.5g/cm³, we have explained the reason in the revised manuscript (page 11 in the revised manuscript):

"In this work, we assume ambient BC mainly composed of primary organic matter with a low degree of oxidation. Based on the study of Nakao et al. (2013), an OC mass density range of 1-1.2 $g/cm^3$ has been used by Liu et al. (2017). $\rho_{BrC}$ = 1.1 $g/cm^3$ is assumed in this work. For the BC mass density, the study of Horvath (1993) gives values of $\rho_{BC}$ = 0.625 $g/cm^3$ and $\rho$ = 1.125 $g/cm^3$. However, Fuller et al. (1999a) pointed out that the values may be not representative for BC in the atmosphere. Medalia and Richards (1972) and Janzen (1980) suggested $\rho_{BC}$ in the range of 1.8-1.9 $g/cm^3$, while Bergstrom (1972) found that the $\rho_{BC}$ value of 1.9-2.1 $g/cm^3$. Bond and Bergstrom (2006) suggested to use a value of 1.8 $g/cm^3$. Figure S8 compares the computations with measurements by assuming $\rho_{BC}$ = 1.8 $g/cm^{3.}$ We assume that $E_{abs}$ and MAC at λ = 0.7 $um$ do not deviate largely with those at λ = 0.781 $um$. Modeled $E_{abs}$ at λ = 0.7 $um$ agrees well with the measurements. Although $E_{abs}$ at λ = 0.404 $um$ seems to be relatively higher than the measurements, it dose not deviate largely with the measurements. However, modeled MAC is a little smaller than the measured MAC. Similar results were obtained

for bare BC ((Kahnert, 2010b), (Liu and Mishchenko, 2005)) . Therefore, $\rho_{BC}$ = 1.8g/cm$^3$ may be a little high for estimation of MAC.

Bond and Bergstrom (2006) concluded that MAC value of 7.5 ± 1.2 $m^2$/g for bare BC can be assumed at λ = 0.55$um$ by reviewing 21 publications of MAC measurements. However, our calculated MAC of 6.02-6.2 $m^2$/g (see Table 2) at λ =0.532 $um$ lies below the range of MAC values suggested by Bond and Bergstrom (2006). Similar conclusions were drawn by Kahnert (2010b) and Liu and Mishchenko (2005). However, our calculated MAC agrees well with the calculated MAC of 6.0 ± 0.1 $m^2$/g by Kahnert (2010b) at λ = 0.55 $um$. As MAC depends significantly on BC mass density, to agree with measurements, Liu and Mishchenko (2005) used $\rho_{BC}$ = *1.0 g/cm$^3$*. However, as pointed by Kahnert (2010b), the measured MAC and modeled MAC were not at the same wavelength, therefore leading to too low retrived $\rho_{BC}$. To raise the computed MAC values to the average observed value of MAC = (7.5 ± 1.2) $m^2$/g, $\rho_{BC}$ = 1.3-1.4 $g/cm^3$ was suggested by Kahnert (2010b). However, this $\rho_{BC}$ value is rather drastic smaller than the value suggested by Bond and Bergstrom (2006). Therefore, Kahnert (2010b) suggested to assume $\rho_{BC}$ = 1.5 ∼ 1.7 $g/cm^3$ to raise the computational MAC results to the lower bound of the observations. By assuming $\rho_{BC}$ = 1.5 $g/cm^3$, the comparison of modeled MAC and $E_{abs}$ with measurements is shown in Figure S9. Overall, the modeled MAC and $E_{abs}$ agree relatively well with the measurement by assuming $\rho_{BC}$ = 1.5 $g/cm^3$. Therefore, $\rho_{BC}$ = 1.5 $g/cm^3$ is assumed in this study."

Some $E_{abs}$ values (eg. $E_{abs}$=5.4, see Figure 11 in the revised manuscript) are scarcely observed in the atmosphere. The most likely reason is the results for BC with thicker BrC coating is unavailable at ultraviolet wavelength. For sensitivity analysis, the BC volume fraction is independent on the BC size. Therefore, to gain $E_{abs}$=5.4, the BC volume fraction should be 5% for all BC. However, in the atmosphere, not all BC is thickly coated. Limited measurements were conducted at ultraviolet wavelength, therefore the measurements are not available for all circumstance. In fact, our calculated results are in general agreement with the measurements in the visible wavelengths. The $E_{abs}$ can reach approximately 3.96 at λ =532$nm$ when $f_{BC}$=5% in this work, which is consistent with the reported $E_{abs}$ value of 2.6-4.0 at Beijing for  λ =470$nm$, China (Xu et al., 2016).

[Figure]

Figure 4 (Figure S8 in the revised manuscript) Comparison of modeled $E_{abs}$ and MAC with measurements, BC mass density is assumed to be 1.8g/cm3, and the measured results are derived from the study of Liu et al. (2015).

[Figure]

Figure 5 (Figure S9 in the revised manuscript) Comparison of modeled Eabs and MAC with measurements, BC mass density is assumed to be 1.5g/cm3, and the measured results are derived from the study of Liu et al. (2015).

Table 1 (Table 2 in the revised manuscript) MAC ($m^2/g$) for bare BC at different $D_f$ ($r_g$ = 0.06 um, $\sigma_g$ = 1.5)

| $\lambda(nm)$ | $D_f$=1.8 | $D_f$=2.2 | $D_f$=2.6 |
|---|---|---|---|
| 350 | 9.30 | 9.03 | 8.48 |
| 404 | 8.14 | 7/95 | 7.60 |
| 532 | 6.20 | 6.11 | 6.02 |
| 700 | 4.68 | 4.64 | 4.65 |

4. In the abstract, please define the '$C_{abs}$', '$K_{BrC}$' before use them.

**Response:** Thanks for pointing it out. We have corrected it in the revised manuscript.

5. In Figure 6, 7 and 11, the range of color bar is suggested to be unified.

**Response:** Thanks for your suggestions. The range of color bar has been unified.

**References**
Adachi, K., Chung, S. H., and Buseck, P. R.: Shapes of soot aerosol particles and implications for their effects on climate, J Geophys Res-Atmos, 115, 2010.
Bergstrom, R. W.: Predictions of the spectral absorption and extinction coefficients of an urban air pollution aerosol model, Atmospheric Environment (1967), 6, 247 – 258, 1972.
Bond, T. C. and Bergstrom, R. W.: Light absorption by carbonaceous particles: An investigative review, Aerosol Science and Technology, 40, 27–67, 2006.
China, S., Mazzoleni, C., Gorkowski, K., Aiken, A. C., and Dubey, M. K.: Morphology and mixing state of individual freshly emitted wildfire carbonaceous particles, Nat Commun, 4, 2013.
Fuller, K. A., Malm, W. C., and Kreidenweis, S. M.: Effects of mixing on extinction by carbonaceous particles, Journal of Geophysical Research-Atmospheres, 104, 15 941–15 954, 1999a.
Horvath, H.: Atmospheric Light-Absorption - a Review, Atmospheric Environment Part a-General Topics, 27, 293–317, 1993.
Janzen, J.: Extinction of Light by Highly Nonspherical Strongly Absorbing Colloidal Particles - Spectrophotometric Determination of Volume Distributions for Carbon-Blacks, Applied Optics, 19, 2977–2985, 1980.
Kahnert, M.: On the Discrepancy between Modeled and Measured Mass Absorption Cross Sections of Light Absorbing Carbon Aerosols, Aerosol Science and Technology,

44, 453–460, <GotoISI>://WOS:000277436300006, 2010b.

Lewis, K. A., Arnott, W. P., Moosmuller, H., Chakrabarty, R. K., Carrico, C. M., Kreidenweis, S. M., Day, D. E., Malm, W. C., Laskin, A., Jimenez, J. L., Ulbrich, I. M., Huffman, J. A., Onasch, T. B., Trimborn, A., Liu, L., and Mishchenko, M. I.: Reduction in biomass burning aerosol light absorption upon humidification: roles of inorganically-induced hygroscopicity, particle collapse, and photoacoustic heat and mass transfer, Atmos Chem Phys, 9, 8949-8966, 2009.

Liu, D. T., Whitehead, J., Alfarra, M. R., Reyes-Villegas, E., Spracklen, D. V., Reddington, C. L., Kong, S. F., Williams, P. I., Ting, Y. C., Haslett, S., Taylor, J. W., Flynn, M. J., Morgan, W. T., McFiggans, G., Coe, H., and Allan, J. D.: Black-carbon absorption enhancement in the atmosphere determined by particle mixing state, Nature Geoscience, 10, 184–U132, 2017.

Liu, L. and Mishchenko, M. I.: Effects of aggregation on scattering and radiative properties of soot aerosols, Journal of Geophysical ResearchAtmospheres, 110, 2005.

Medalia, A. I. and Richards, L. W.: Tinting Strength of Carbon-Black, Journal of Colloid and Interface Science, 40, 233–&, 1972.

Nakao, S., Tang, P., Tang, X. C., Clark, C. H., Qi, L., Seo, E., Asa-Awuku, A., and Cocker, D.: Density and elemental ratios of secondary organic aerosol: Application of a density prediction method, Atmospheric Environment, 68, 273–277, 2013.

Wang, Y. Y., Liu, F. S., He, C. L., Bi, L., Cheng, T. H., Wang, Z. L., Zhang, H., Zhang, X. Y., Shi, Z. B., and Li, W. J.: Fractal Dimensions and Mixing Structures of Soot Particles during Atmospheric Processing, Environ Sci Tech Let, 4, 487-493, 2017.

Xu, X. Z., Zhao, W. X., Zhang, Q. L., Wang, S., Fang, B., Chen, W. D., Venables, D. S., Wang, X. F., Pu, W., Wang, X., Gao, X. M., and Zhang, W. J.: Optical properties of atmospheric fine particles near Beijing during the HOPE-J(3)A campaign, Atmos Chem Phys, 16, 6421-6439, 2016.

---

## Author Comment (AC3) · 5 Oct 2018

**Response to the comments of the Reviewer #3**

(The responses are highlighted in blue)

First of all, we would like to thank the three anonymous reviewers for their thoughtful review and valuable comments to the manuscript. In the revision, we have accommodated all the suggested changes into consideration and revised the manuscript accordingly. All changes are highlighted in the revised manuscript in **BLUE** in the revision.

In this response, the questions and comments of reviewers are in black font, and responses are highlighted in **BLUE**. The changes made in the revised manuscript are marked in RED font.

The topic of black carbon (BC) absorption enhancement has been investigated by numerous previous modeling/lab/field studies. The present manuscript systematically quantified the effects of brown carbon coating and associated morphological properties on BC absorption enhancement and proposed a "sunglasses effect", which provides some new understanding in this topic. This study is suitable for ACP and its structure is clear. Before it can be considered for publication, I have a few comments and suggestions to help improve the manuscript.

Abstract: The authors mentioned "thickly-coated" and "thinly-coated" here. How thick is "thickly-coated"? Please quantitatively define it here and in the main text as well

.**Response:** Thanks for your comments and valuable suggestions. In this work, BC is defined as thickly coated when the BC volume fraction is lower than 20%, and other BC is considered to be thickly coated. We have defined it in the abstract and the introduction of the revised manuscript.

Abstract (Lines 11-12): "the uncertainties … have differences of less than 2.6% and 6% …". The expression "uncertainties have differences …" is weird. Please rephrase this sentence.

**Response:** Thanks for your comments. We agree that this sentence would make readers confused. After carefully consideration, we think that this point is not the most important in this work, therefore we have removed this sentence in the revised manuscript, and the abstract is rewritten. They are all marked in blue in the revised manuscript.

Page 1, Line 20: "second contribution" should be "second contributor". The reference for this sentence should also include Bond et al., 2013 (JGR).

**Response:** Thanks for pointing it out. We have added the reference in the revised manuscript.

Introduction: The authors mentioned that BC absorption enhancement varies significantly in different field measurements, which could be due to complex morphology and mixing state during BC aging processes. But one missing part is the evidence for the complex BC morphology and mixing state observed in field measurements (e.g., Y. Wang et al. 2017, doi:10.1021/acs.estlett.7b00418; S. China et al. 2013, doi:10.1038/ncomms3122). I suggest including several sentences in the introduction to point out this aspect.

**Response:** Thanks for your comments and suggestions. We have added it in the introduction of the revised manuscript:

"Freshly emitted BC commonly presents fractal structures. As the BC ages in the atmosphere, BC becomes more compact and OC materials can condense onto the particles. Therefore, BC can be embedded in an OC shell (China et al., 2013a; Wang et al., 2017). When non-BC fraction is low, BC can still present near fractal structure (referred as thinly coated in this study) (Wang et al., 2017). As BC further coated, BC aggregates are collapsed into more compact and spherical clusters when fully engulfed in coating material (referred as thickly-coated BC in this study) (Zhang et al., 2008b)."

Page 4, Lines 8-10: Using $D_f$ values to define "thickly-coated" and "thinly-coated" BC is not straightforward. Why not use the coating thickness or mass directly? There may be some situations where $D_f$ is smaller than 2.6, but the coating is still more than that of BC with a relatively higher $D_f$.

Response: We are sorry for not clarifying the definition of "thickly-coated" and "thinly-coated" BC. We don't define "thickly-coated" and "thinly-coated" BC using $D_f$ values. In this work, BC is defined as thickly coated when the BC volume fraction is lower than 0.2.

We do acknowledge that in some cases, $D_f$ of BC is small but the coating fraction is high (such as the partially encapsulated coated BC). In general, however, thickly-coated BC is with compact structures due to squeeze effect. China et al. (2013) reported that atmospheric soot particles with the thickest coating had the highest fractal dimension. Compact structures are commonly observed for aged BC (Moffet and Prather, 2009). Therefore, $D_f$=2.6 was assumed for thickly-coated BC. This aging process is also assumed in other studies (such as the study of Wu el at., 2016; 2018 and Kanngiesser et al., 2018).

Page 5, Lines 6-10: Recently, another important and efficient particle light-scattering method, the geometric-optics surface-wave (GOS) method (Liou et al. 2011, doi.org/10.1016/j.jqsrt.2011.03.007; C. He et al. 2016, doi.org/10.1016/j.jqsrt.2016.08.004), has also been developed and applied to resolve complex BC coating structures and showed consistent results with MSTM, which could be included here

Response: Thanks for your comments. We have included it in the revised manuscript.

Page 6, Lines 13-14: The authors assumed BrC coatings are uniformly distributed over the BC surface, but they also argued that the blocking effect of coating is important, which could be affected by how coating materials are distributed over BC particle surface. Thus, assuming the uniform distribution of BrC coating may lead to nontrivial biases in calculations. Could the authors comment or add some discussions on this?

Response: We are sorry for not clarifying the means of the sentence in Page 6, Lines 13-14, and it doubts you. What we really want to express is that the composition ratio of BC to coating is independent to size. It is reasonable to make the simplification for easily understanding the effects of brown coatings.

The coatings are believed to be uniformly distributed over the BC surface for thinly coated BC (closed cell). Moreover, Kahnert (2017) has demonstrated the differences between closed cell model and more morphologically realistic model are not large for calculation of absorption of thinly-coated BC. For thickly coated BC, the relative position of BC to coatings can also affect the absorption, while one should not expect large difference in absorption (Liu et al. 2017, He et al. 2015). In this work, the geometric center of black carbon are located in the center of BrC sphere.

Page 7, Lines 11-12: "Generally, $E_{abs}$ increases … with increasing $k_{BrC}$." Is this true for all wavelengths? Please clarify here.

Response: Thanks for your comments. It is true that $E_{abs}$ increases with increasing $k_{BrC}$ for all wavelengths selected in this work. The reason is that the absorption of BrC increases with $k_{BrC}$, which contributes the increase of $E_{abs}$. Even though the sunglass effect also increase with $k_{BrC}$, the increase in sunglass effects is relative small, compared with the increase caused by BrC absorption, therefore leads to increase in $E_{abs}$.

Page 7, Lines 16-17: "… compared with BC with non-absorbing coatings, $E_{abs}$ for thinly-coated BC

with absorbing coatings seems to be less wavelength-dependent, …" This is interesting but a little counter-intuitive. Could the authors provide some explanations?

**Response:** Thanks for your comments. The wavelength-dependence of $E_{abs}$ for absorbing coatings is at fixed $k_{BrC}$. The results is the interaction of the lensing effect and BrC absorption. At fixed $k_{BrC}$, lensing effect commonly decrease with wavelength for thinly-coated BC, while BrC absorption can increase with wavelength at fixed $k_{BrC}$. Therefore, the two effects are counteracted for thinly-coated BC. Therefore, $E_{abs}$ wavelength–dependences of thinly-coated BC decrease with wavelength. However, for thickly coated BC, the lensing effect also increases with wavelength. Therefore, the effect of lensing effect and BrC absorption are superimposed, which leads to larger wavelength-dependence of $E_{abs}$.

Section 3: The authors highlighted two important but opposite effects: conventional lensing effect and sunglasses effect. It is interesting to see how these two effects vary with $k_{BrC}$, $D_f$, and wavelength. Since the authors already calculated the absorption due to these two effects, it is straightforward to calculate the contributions of these two effects to the total absorption enhancement. This would be very informative and worth discussing. Also, according to the authors' arguments, there should be one critical point (or critical $k_{BrC}$ value) for the two effects to be the same. It would be very interesting to see what this point/value is.

**Response:** Thanks for your comments. We agree that it is straightforward to calculate the contributions of these two effects to the total absorption enhancement.

After carefully consideration, we think lensing effects is for non-absorbing coatings, the definition is as follows:

$$E_{len\sin g} = \frac{C_{abs\_coated\_non\text{-}absorbing}}{C_{abs\_bare}}$$

Combining the comments of other reviewers, we think that the lensing effect defined in Equation (5) of the previous version of the manuscript is a little misleading. The lensing effect defined in Equation (5) of previous version is the comparison of the absorption of BC coated by BrC with an external mixture of BrC and BC. It is resulted from the interaction of lensing effect and sunglass effect.

Liu et al. (2017a) defined the lensing effect as the absorption enhanced by addition of non-black carbon. However, from a physical point of view, for BC with BrC coatings, the definition may be

not clear, and it can be confused with $E_{abs}$. Therefore, we redefined the lensing effect as the absorption enhanced by addition of non-absorbing coatings in the revised manuscript. In addition, we assume that the lensing effect of BC with absorbing coatings is the same as those with non-absorbing coatings. We believe this is a reasonable assumption since the BrC and nonabsorbing coating have a similar value of real part of refractive index..

We clearly defined the sunglass effect in the revised manuscript. We contribute $E_{abs}$ of BC with BrC coatings to lensing effect, BrC absorption enhancement and sunglass effect. Therefore:

$$E_{Sunglass} = -\frac{C_{abs\_coated} - C_{abs\_BrC\_shell} - C_{abs\_non-absorbing}}{C_{abs\_bare}}$$

The negative sign represents that the sunglass effect can cause the decrease of total absorption. There is indeed be one critical point (or critical $k_{BrC}$ value) for the two effects to be the same ($E_{abs\_internal}<1$ in the revised manuscript). However, the critical $k_{BrC}$ value is dependent on the mixing states and particle size. Therefore, it is difficult to give the critical $k_{BrC}$ value for each case. Nevertheless, we have investigated the effects of mixing states and particle size in the revised manuscript:

" Generally, the threshold $k_{BrC}$ value decreases with particle size and coatings thickness, as $E_{abs\_internal}$ of BC thickly-coated with absorbing coatings decreases with particle size and coatings thickness (see Figure 6 and Figure 9)."

Page 8, Line 19: "shorter wavelength". Please give a more quantitative wavelength range

**Response**: Thanks for your comments. This sentence has been corrected in to "larger absorption enhancement can be observed by increasing λ from ultraviolet region to visible region".

Page 8, Line 15 (and elsewhere): "relative errors". I suggest using "relative uncertainty" instead of "error".

**Response:** Thanks for your suggestions. We have corrected it in the revised manuscript.

Page 12, Line 1: "combined of $E_{abs}$ …". Should it be "combining $E_{abs}$ …"?

**Response:** Thanks for your suggestions. We have corrected it in the revised manuscript.

Section 4: Could the authors add some discussions on how to apply their results in this study to

climate models? Current climate models do not simulate any morphological information of aerosols and generally assume a core-shell structure or external mixing for aerosols.

**Response:** Thanks for your comments. In the revised manuscript, we have added the calculation of mass absorption cross section (MAC). The MAC of bare BC can be estimated by measurements or parameterization. After investigating the mechanism of absorption enhancement, we can understand and simulated the MAC for coated BC in various circumstances. If validated by measurements, we can incorporated the results into the CTMs. We have added some discussions on this aspect in the section 4:

"In this work, complex morphologies and mixing states are considered. Although current climate models do not simulate any morphological information of aerosols, many laboratory studies has been conducted to investigate the BC morphologies in different mixing states and in different regions. Therefore, our calculations can be applied according to specific mixing states (such as composition ratios) and regions. However, we acknowledge that the understanding of the relation between BC morphology and the composition ratio is still limited. Therefore, further laboratory investigations for the coated BC morphologies should be conducted in the future."

**Reference**

China, S., Mazzoleni, C., Gorkowski, K., Aiken, A. C., and Dubey, M. K.: Morphology and mixing state of individual freshly emitted wildfire carbonaceous particles, Nat Commun, 4, 2013.

Cheng, T. H., Wu, Y., and Chen, H.: Effects of morphology on the radiative properties of internally mixed light absorbing carbon aerosols with different aging status, Opt Express, 22, 15904-15917, 2014.

He, C., Liou, K. N., Takano, Y., Zhang, R., Zamora, M. L., Yang, P., Li, Q., and Leung, L. R.: Variation of the radiative properties during black carbon aging: theoretical and experimental intercomparison, Atmos Chem Phys, 15, 11967-11980, 2015.

Kahnert, M.: Optical properties of black carbon aerosols encapsulated in a shell of sulfate: comparison of the closed cell model with a coated aggregate model, Opt Express, 25, 24579-24593, 2017.

Kanngiesser, F., and Kahnert, M.: Calculation of optical properties of light-absorbing carbon with weakly absorbing coating: A model with tunable transition from film-coating to spherical-shell coating, J Quant Spectrosc Ra, 216, 17-36, 2018.

Liu, C., Li, J., Yin, Y., Zhu, B., and Feng, Q.: Optical properties of black carbon aggregates with non-absorptive coating, J Quant Spectrosc Ra, 187, 443-452, 2017.

Wu, Y., Cheng, T. H., Zheng, L. J., and Chen, H.: Black carbon radiative forcing at TOA decreased during aging, Sci Rep-Uk, 6, 2016.

Wu, Y., Cheng, T. H., Liu, D. T., Allan, J. D., Zheng, L. J., and Chen, H.: Light Absorption Enhancement of Black Carbon Aerosol Constrained by Particle Morphology, Environ Sci Technol, 52, 6912-6919, 2018.

---

## Author Response (AR2)

**Responses to the comments of the reviewers**

(The responses are highlighted in blue)

First of all, we would like to thank the three anonymous reviewers for their thoughtful review and valuable comments to the manuscript. In the revision, we have accommodated all the suggested changes into consideration and revised the manuscript accordingly. All changes are highlighted in the revised manuscript in **BLUE** in the revision.

In this response, the questions and comments of reviewers are in black font, and responses are highlighted in **BLUE**.

*Response to the co-editor*

See comments by reviewer. I'm hoping these should be very simple to address.

**Response:** Thanks for your comments. We have addressed the comments as follows.

*Response to the reviewer #2*

The revised manuscript is clearer and more compelling, and I recommend this paper for publication after minor revise.

**Response:** Thanks for your comments. We are glad to know that you are satisfactory with our most replies.

In 2nd paragraph of page 2, the absorption enhancement is usually investigated at 532nm, thus it is suggested to show the effects at this wavelength as well.

**Response:** Thanks for your suggestions. Actually, we investigated these effects at 4 typical wavelengths, including 532nm. Please see the figures 2-12 in the previous version of our manuscript.

In Figure 11, $E_{abs}$ may be larger than 5.4 at 350nm. The previous measurements and simulations indicated that $E_{abs}$ is increased to a stable value for large $M_R$ (Mikhailov et al., Peng et al., Wu et al.), thus, more simulations would be necessary for the peak $E_{abs}$ value.

References:

Mikhailov, E. F., et al. Optical properties of soot-water drop agglomerates: An experimental study. J. Geophys. Res. 2006, 111, 1576-1585.

Peng, J., et al. Markedly enhanced absorption and direct radiative forcing of black carbon under polluted urban environments. Proc. Natl. Acad. Sci. U. S. A. 2016, 113, 4266−4271.

Wu, Y., et al. Light absorption enhancement of black carbon aerosol constrained by particle morphology. Environ. Sci. Technol. 2018, 52, 6912-6919.

**Response:** Thanks for your comments. $E_{abs}$ may be indeed larger than 5.4 at 350nm when $M_R$ is larger. Therefore, in the revised manuscript, we have clarified that $E_{abs}$=5.4 was obtained at $M_R$=13.9.

In the study of Wu et al. (2018), they considered $M_R$ of reaching approximately 100, while $M_R$ of most coated BC is below 10 (Liu et al., 2017) (see the figure 1a of Liu et al. (2017)). According to previous studies, the shell/core ratio $D_p/D_c$ (spherical equivalent particle diameter divided by BC core diameter) was observed to be commonly in the range of 1.1-2.7 (Liu et al., 2015) (Zhang et al., 2016), and the corresponding $M_R$ is approximately 0.24-13.9. Therefore, $M_R$ of 0-13.9 is considered in this work. We have added some descriptions in the revised manuscript on this aspect.

When BC is coated with non-absorbing materials, $E_{abs}$ can indeed reach a stable value, and such value was also observed in this work (see figure 11 of our manuscript for $E_{abs}$ at 0.7 um wavelength). However, when BC is coated with absorbing coatings, $E_{abs}$ will always increase with increasing $M_R$, and will not reach a stable value. We believe that the absorbing coatings will always promote the total absorption as coatings increases. Even though the sunglass effect also increase with the coating thickness, the increase of sunglass effect is relative small compared with the increased BrC absorption. In the extreme case, the $M_R$ is big enough so that the absorption of BC core can be negligible. Therefore, we do not conduct more simulations for the peak $E_{abs}$ value, as the peak $E_{abs}$ value does not exist for BC coated with absorbing coatings.

References

Liu, D., Whitehead, J., Alfarra, M. R., Reyes-Villegas, E., Spracklen, Dominick V., Reddington, Carly L., Kong, S., Williams, Paul I., Ting, Y.-C., Haslett, S., Taylor, Jonathan W., Flynn, Michael J., Morgan, William T., McFiggans, G., Coe, H., and Allan, James 
[revised manuscript text omitted]

---

## Author Response (AR3)

In this response, the questions and comments of reviewers are in black font, and responses are highlighted in **BLUE**, and the modifications in the revised manuscript are marked in **RED**.

I think perhaps the authors have misunderstood the reviewer's comments; there is nothing incorrect in what is shown, however there are minor issues of emphasis in what is presented. However, as these modifications are relatively trivial and do not impact on the overall science of what is being presented, I have marked these as "technical corrections" that the authors should address before final submission.

**Response:** Thanks for your comments. We did misunderstand the reviewer's comments, and we have addressed the reviewer's comments in the revised manuscript.

1. The point that I think the reviewer is making concerning 532nm isn't that the work should take place (which it clearly has), but that it may make sense to refer to the $E_{abs}$ values at this wavelength more, specifically in the abstract and conclusions, for the sake of comparability with the previous literature and relevance to the peak in the solar spectrum. Currently the main emphasis is placed on 350nm, but I would suggest also referring to 532nm

**Response:** Thanks for your suggestions. We have presented the $E_{abs}$ value at $532nm$ in the abstract and conclusion of the revised manuscript:

"Specifically, as $M_R$ increase to approximately 13.9, $E_{abs}$ of larger than 3.96 can be observed at 0.532µm, which is a little higher than the commonly measured $E_{abs}$ of 1.05 ∼ 3.5 at this wavelength."

2. Regarding the reviewer's point about the $E_{abs}$ value of 5.4, the way the abstract is currently worded ("$E_{abs}$ can reach approximately 5.4"), this implies that this is a maximum value, however an inspection of figure 11 would indicate that even larger $E_{abs}$ values could be reached if greater $M_R$ values had been modelled. I am not suggesting that further modelling take place, but it should be made clearer, particularly in the abstract, that the value could be even greater than 5.4 at higher MR values.

**Response:** Thanks for your suggestions. We have clarified this aspect in the abstract of the revised manuscript:

[revised manuscript text omitted]

There is a different dependence on $M_R$ for $E_{abs\_internal}$ at different wavelengths. $E_{abs\_internal}$ increases with $M_R$ at relative long wavelengths (eg. $\lambda$ =0.7 [..[74] ]$\mu m$), while decreases as the coatings become thicker at relative short wavelengths (0.404 [..[75] ]$\mu m$ and 0.532 [..[76] ]$\mu m$). This phenomenon can also be explained from physical insights. When the wavelength
* * *
[62]removed: $um$
[63]removed: $um$
[64]removed: $um$
[65]removed: $um$
[66]removed: $um$
[67]removed: $um$
[68]removed: $um$
[69]removed: $um$
[70]removed: $um$
[71]removed: $um$
[72]removed: $um$
[73]removed: $um$
[74]removed: $um$
[75]removed: $um$
[76]removed: $um$

is short, increased thickness of the coatings may lead to a greater sunglass effect, which weakens the total absorption of the coatings and BC. However, at $\lambda = 0.7$ [..[77] ]$\mu m$, enhanced $E_{abs\_internal}$ can be obtained by increasing the coatings due to the negligible blocking effects of the coatings. In addition, $E_{abs\_internal}$ increases with wavelength due to the decrease in coating absorption (see Figure 2). $E_{abs\_internal}$ of thickly-coated BC is insensitive to $M_R$ at $\lambda = 0.35$ $\mu m$ due to the similar variations

5    of $E_{abs\_lensing}$ and $E_{Sunglass}$ with $M_R$. As $M_R$ varies from 6.6 to 13.9, $E_{abs\_lesnig}$ increases from 2.204 to 2.363, 2.214 to 2.390, 2.216 to 2.473, 2.165 to 2.509 at $\lambda = 0.35$ [..[78] ]$\mu m$, 0.404 [..[79] ]$\mu m$, 0.532 [..[80] ]$\mu m$, and 0.7 [..[81] ]$\mu m$, respectively. While $E_{Sunglass}$ largely affected by wavelengths. At $\lambda = 0.35$ [..[82] ]$\mu m$, $E_{Sunglass}$ is in the range from 1.523 to 1.807, while $E_{Sunglass}$ approaches 0 at $\lambda = 0.7$ [..[83] ]$\mu m$. It is also can be seen from Figure 11 that $E_{Sunglass} > E_{abs\_lensing} - 1$ at $\lambda = 0.35$ [..[84] ]$\mu m$ and 0.404 [..[85] ]$\mu m$. Therefore, $E_{abs\_internal}$ is less than 1.

10      You et al. (2016) demonstrated that there are different wavelength dependencies for BC that is coated with absorbing and weak absorbing materials. $E_{abs}$ for BC coated with humic acid was observed to vary from 3.0 to approximately 1.6 as $\lambda$ increased from 0.554 [..[86] ]$\mu m$ to 0.84 [..[87] ]$\mu m$, while it seemed to be essentially wavelength-independent for BC that is coated with sodium chloride. Figure 12 compares the wavelength dependencies of BC coated with non-absorbing materials and BrC. For thinly-coated BC, there are substantial wavelength dependencies for BC coated with BrC. By setting $f_{BC}$ to be

15    40%, $E_{abs}$ increases from 1.15 to 1.57 with $\lambda$ varying from 0.7 [..[88] ]$\mu m$ to 0.35 [..[89] ]$\mu m$, which results in approximately 49.6% increase. However, when coated with non-absorbing materials, $E_{abs}$ exhibits small wavelength-dependences. This leads to approximate 8.7% increases as $\lambda$ decreases from 0.7 [..[90] ]$\mu m$ to 0.35 [..[91] ]$\mu m$. Furthermore, for thickly-coated BC, $E_{abs}$ is significantly wavelength-dependent for BC with BrC coatings. The decrease in $\lambda$ from 0.7 [..[92] ]$\mu m$ to 0.35 [..[93] ]$\mu m$ would result in approximate 100% increase in $E_{abs}$, while $E_{abs}$ seems to be essentially wavelength-independent for BC with non-

[revised manuscript text omitted]
 ($M_R$) is over 13.9, $E_{abs}$ can exceed 5.4 for BC with brown coatings at $\lambda = 0.35$ μm under a typical size distribution."

As a further technical point, I note that the authors have been denoting microns as 'um', but it is far more professional to use "μm". The authors should change this throughout before submitting their final version.

**Response:** Thanks for pointing our mistake out. We have modified it in the revised manuscript.